# Unveiling the Entropy Dynamics of Chain-of-Thought Reasoning

Ting Xu [1]  Xu He [2]  Yupu Lu [3]  Jiankai Sun [1]  Dong Li [2]  Wai Lam [1]  Jianye Hao [2]

## Abstract

This paper investigates the entropy dynamics of Chain-of-Thought (CoT) and uncovers a consistent two-phase structure: an *Uncertainty Region* of exploration transitioning sharply to a *Confidence Region* of convergence. We demonstrate that the Confidence Region possesses two critical properties: 1) *High Reliability*—answers in the confidence region become highly accurate and stable, and 2) *High Redundancy*—models generate unnecessary tokens long after reaching the correct answer. These properties unlock more efficient and reliable inference strategies: 1) *Early Exit* leverages reliability and redundancy to terminate computation safely when returns diminish, and 2) *Test-Time Scaling* uses the Confidence Region signal to prioritize converged trajectories. To operationalize these insights, we formulate Confidence Region detection as a sequential change-point detection problem, being the first to apply classical change-point methods to monitor CoT reasoning. Using the Cumulative Sum (CUSUM) algorithm, a statistically optimal change-point detector, we develop a training-free framework for real-time inference control. Experiments show our approach establishes a superior Pareto-frontier for early exit. CUSUM achieves 63.06% accuracy with 11.1% token reduction, outperforming DEER and Dynasor by 3.28% and 4.36% in accuracy respectively. For test-time scaling, CUSUM-weighted voting consistently outperforms self-consistency.

## 1. Introduction

Chain-of-Thought (CoT) reasoning (Wei et al., 2022) has unlocked remarkable capabilities in Large Language Models (LLMs), enabling them to solve complex tasks in mathematics, science, and coding (Shao et al., 2024; Guo et al., 2025; Merrill & Sabharwal, 2024; Bai et al., 2025; Anil et al., 2023). Despite this empirical success, understanding how CoT reasoning unfolds remains an open question. Recent research has approached this through theoretical expressiveness, internal mechanisms, and semantic transitions (Liu et al., 2024; Dutta et al., 2024; Li et al., 2025; Yu et al., 2025). These studies treat the CoT reasoning as discrete segments and focus on understanding what happens within individual steps, layers, or local state transitions—offering insights into the local properties of reasoning.

However, reasoning is not merely a sequence of independent steps—it is a continuous process that evolves from initial exploration toward a final solution (Yao et al., 2023). While existing work provides insights into local transitions and components, understanding the *global dynamics* of this process remains a challenge: *How does reasoning progress from exploration to convergence?* Answering this question is fundamental to deciphering model reasoning—distinguishing whether a solution emerges through gradual refinement or via a sudden "Aha" moment (Yang et al., 2025b).

In response, we view LLM reasoning as a process of uncertainty reduction, a perspective shared by Wu et al. (2025). In this framework, CoT reasoning steps serve as incremental information gains; each step reduces ambiguity by providing a more structured context for subsequent predictions (Da et al., 2025; Wu et al., 2025). To quantify this uncertainty reduction, we track the model's confidence in its final answer at each reasoning step through predictive entropy—a measure that captures the model's distributional uncertainty over possible answers. Specifically, high predictive entropy denotes active exploration among competing hypotheses, while a low predictive entropy indicates a deterministic convergence (Wang et al., 2025a; Zhang et al., 2025). Thus, the predictive entropy dynamics allow us to pinpoint the exact evolution of reasoning. To the best of our knowledge, this is the first systematic analysis of entropy dynamics throughout the CoT generation process.

We discover that CoT reasoning follows a two-phase structure rather than a gradual progression: a high-entropy *Uncertainty Region* of stochastic exploration followed by an abrupt transition into a low-entropy *Confidence Region*

[1]The Chinese University of Hong Kong [2]Huawei Technologies Ltd [3]School of Computing and Data Science, The University of Hong Kong. Correspondence to: Wai Lam <wlam@se.cuhk.edu.hk>.

*Proceedings of the 43rd International Conference on Machine Learning*, Seoul, South Korea. PMLR 306, 2026. Copyright 2026 by the author(s).

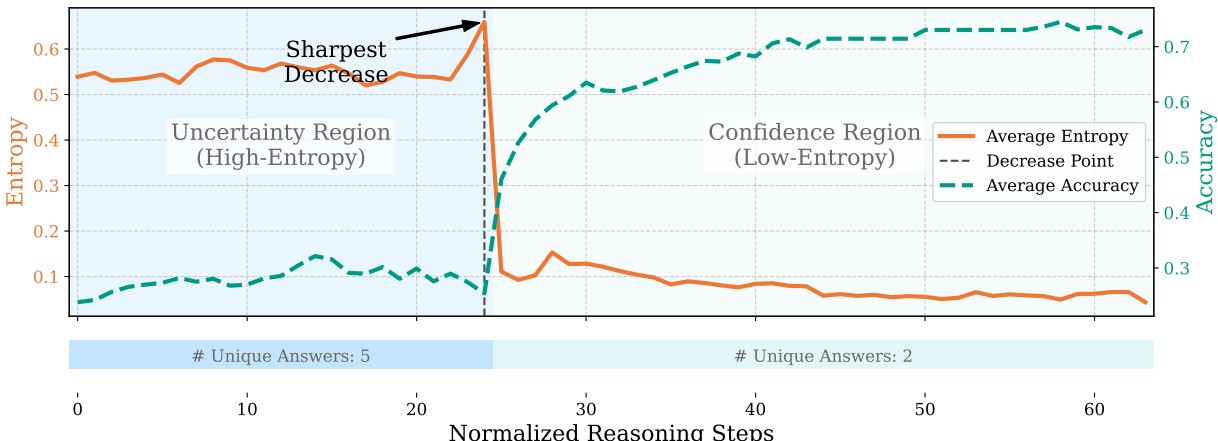

*Figure 1.* **Dynamics of Entropy and Accuracy on Qwen3-4B-Thinking-2507**: CoT reasoning exhibits a two-phase structure: (1) an Uncertainty Region, where high entropy and diverse answers reflect exploration of multiple logical paths, and (2) a Confidence Region, where entropy collapse and answer stabilization signal convergence toward a reliable solution.

of deterministic convergence. Crucially, the Confidence Region exhibits two key properties that make it a prime candidate for inference optimization: *High Reliability*—answers become highly accurate and stable upon entering this region—and *High Redundancy*—models continue generating redundant tokens well after reaching the correct answer. Together, these findings answer our opening question: reasoning evolves by abruptly shifting from stochastic exploration to deterministic convergence. After the model enters the Confidence Region, the reasoning is largely complete; continued generation typically yields diminishing returns.

These properties unlock two efficient and reliable inference strategies: *Early Exit* leverages reliability for safe termination while exploiting redundancy to prune unnecessary computation, and *Test-Time Scaling* leverages reliability to distinguish converged trajectories from those still in exploration. Both applications require real-time detection of when this transition occurs.

We formalize this detection task as a *sequential change-point detection* problem, bridging CoT dynamics with classical sequential analysis theory (Berger, 2008). To the best of our knowledge, we are the first to apply classical change-point detection methods to monitor CoT reasoning. Specifically, we adopt the Cumulative Sum (CUSUM) algorithm (Lorden, 1971), a statistically optimal method that recursively aggregates log-likelihood ratios to monitor evidence of convergence. This training-free approach enables efficient early exit and test-time scaling by reliably identifying the Confidence Region transition in real-time.

We evaluate our method in a diverse suite of models—DeepSeek-R1-Distill-Qwen-7B, Qwen3-4B-Thinking-2507, and Qwen3-14B—on rigorous reasoning benchmarks, including AIME24, AIME25, and GPQA-Diamond. In early-exit scenarios, our approach consistently

outperforms competitive baselines including DEER (Yang et al., 2025a) and Dynasor (Fu et al., 2025a), achieving superior Pareto frontiers across all models. It achieves an average accuracy of 63.06% with 11.1% token reduction, outperforming DEER (59.78%) and Dynasor (58.70%) by 3.28% and 4.36% in accuracy respectively. In test-time scaling, CUSUM-weighted voting consistently surpasses standard self-consistency, with the performance gap widening as the number of sampled trajectories increases. This improvement is driven by the final CUSUM score's ability to serve as a reliable quality indicator, as evidenced by the clear statistical separation between the scores of correct and incorrect trajectories. These results demonstrate that by identifying the transition to the confidence region, our method successfully eliminates computational waste and distills high-quality reasoning, establishing a more efficient and reliable method for LLM inference.

## 2. Unveiling Entropy Dynamics

To model the dynamics from exploration to convergence, we use predictive entropy to quantify uncertainty about the answer at each iteration. For an input $X$, the model generates a sequence of reasoning steps $\{T_i | i = 1, 2, ...\}$, where $T_i$ is a sequence of $k$ tokens conditioned on the input and all preceding steps: $T_i \sim \text{LLM}(T_i | X, T_{1:i-1})$. At each step $i$, we obtain an intermediate answer after $T_i$:

$$A_i \sim \text{LLM}(A_i | X, T_{1:i}, \mathcal{I}_{\text{ans}}), \tag{1}$$

where $A_i$ consists of $m_i$ tokens $(a_{i,1}, \cdots a_{i,m_i})$ and $\mathcal{I}_{\text{ans}}$ is an answer-inducing prompt, e.g., `</think> The answer is \boxed`. The predictive entropy of $A_i$ is computed as:

$$\mathcal{H}_i = -\frac{1}{m_i} \sum_{j=1}^{m_i} \sum_{u \in \mathcal{V}} \mathcal{P}_j(u|\cdot) \log \mathcal{P}_j(u|\cdot), \tag{2}$$

where $\mathcal{V}$ is the vocabulary set, and $\mathcal{P}_j(u|\cdot) = \mathcal{P}(u|X, T_{1:i}, \mathcal{I}_{ans}, a_{i,1:j-1})$ is the probability of token $u$ from the LLM. High entropy indicates exploration; low entropy indicates convergence.

**Setup.** To investigate the dynamics of CoT, we analyze predictive entropy $\mathcal{H}_i$ in 100 randomly selected instances from the Bespoke-Stratos-17k dataset (Labs, 2025). This dataset provides a heterogeneous collection of tasks spanning coding, mathematics, science, and logic, ensuring the diversity of our observations. To give a population-level visualization, we align trajectories of varying lengths at their point of sharpest entropy decrease and normalize the steps and entropies via linear interpolation.

**Observation and Formulation.** The aggregated entropy dynamics for Qwen3-4B-Thinking-2507 (Figure 1) reveals a distinct two-phase structure. We identify the *Uncertainty Region*, characterized by high-entropy exploration, followed by an abrupt transition to the *Confidence Region* defined by stable low entropy indicative of solution convergence. We formalize this two-phase structure as a regime shift model. Let $\{\mathcal{H}_i \mid i = 1, 2, ...\}$ denote the sequence of predictive entropies observed at each reasoning step. We model this sequence as switching between two stochastic regimes at a change-point $\nu$:

$$\mathcal{H}_i \sim \begin{cases} f_0 & \text{if } i < \nu \quad \text{(Uncertainty Region)} \\ f_1 & \text{if } i \geq \nu \quad \text{(Confidence Region)} \end{cases} \quad (3)$$

where $f_0$ and $f_1$ are the probability distribution of Uncertainty and Confidence regions, respectively.

To demonstrate the universality of this phenomenon, we provide additional visualizations for DeepSeek-R1-Distill-Qwen-7B and Qwen3-14B in Appendix Figures 7 and 8, as well as individual trajectories in Figure 10. These visualizations confirm that the abrupt transition is a consistent property across different model architectures and scales.

**Characterizing the Transition.** To understand the factors governing this regime shift, we analyze the normalized change-point $\nu_{norm} = \nu/L$ (where $L$ denotes the total trajectory length) across models, difficulty levels. Table 1 summarizes results computed over 100 calibration instances. Three key findings emerge from this analysis. *(1) Problem difficulty governs transition timing.* On easy problems, models enter the Confidence Region within the first third of the trajectory (30–34%); on hard problems, the transition is delayed to the middle-to-late stage (52–56%). This indicates that the *High Redundancy* property identified above is predominantly a phenomenon of easy instances—once the model converges early, subsequent reasoning steps contribute diminishing marginal informa-

*Table 1.* Normalized change-point $\nu_{norm}$ and its Pearson correlation with final accuracy, stratified by model and difficulty level. DeepSeek-7B and Qwen3-4B are shorthand for DeepSeek-R1-Distill-Qwen-7B and Qwen3-4B-Thinking-2507, respectively.

| Model | $\nu_{norm}$ (Easy) | $\nu_{norm}$ (Hard) | Corr($\nu_{norm}$, Acc) |
|---|---|---|---|
| DeepSeek-7B | 33.9% | 56.4% | −0.45 |
| Qwen3-4B | 31.0% | 54.0% | −0.36 |
| Qwen3-14B | 30.0% | 52.0% | −0.26 |

tion. *(2) Stronger models converge earlier.* Across all difficulty levels, stronger models exhibit earlier transition points (Qwen3-14B: 30%/52% vs. DeepSeek-R1-Distill-Qwen-7B: 33.9%/56.4%). This suggests that model capacity directly accelerates the exploration-to-convergence transition — stronger models need fewer reasoning steps to resolve uncertainty. *(3) Earlier convergence correlates with correctness, particularly for distilled models.* The negative correlation between $\nu_{norm}$ and accuracy is strongest for the distilled 7B model ($r = -0.45$) and weakest for the natively trained 14B model ($r = -0.26$). We hypothesize that distilled models possess less robust representations, coupling convergence timing more tightly with reasoning quality.

**Properties and Applications of the Confidence Region.** Figure 1 reveals two key properties of the Confidence Region that enable inference optimization. 1. *High Reliability*: The Uncertainty Region exhibits unreliable exploration with low accuracy ($< 20\%$). Upon entering the Confidence Region, accuracy surges sharply to a high plateau ($> 60\%$), indicating stable reasoning. 2. *High Redundancy*: Although the correct answer typically emerges early in the Confidence Region, models continue generating substantial redundant tokens—often exceeding 30% of the total trajectory length—well beyond solution discovery.

These properties enable two efficient inference strategies. **Early Exit** exploits both reliability (safe termination upon entering the Confidence Region) and redundancy (eliminating wasteful computation, as continued generation yields minimal accuracy gains). **Test-Time Scaling** leverages reliability as a trajectory quality indicator: trajectories that successfully transition to the Confidence Region (finite $\nu$) demonstrate stronger reasoning than those remaining indefinitely in the Uncertainty Region ($\nu = \infty$), enabling principled weighting beyond uniform self-consistency (Wang et al., 2023). Both strategies require real-time detection of the Confidence Region transition.

## 3. Confidence Region Detection

We formalize the detection of the Confidence Region—identifying the unknown transition time $\nu$—as a sequential change-point detection problem, grounded in sequential analysis theory (Lorden, 1971).

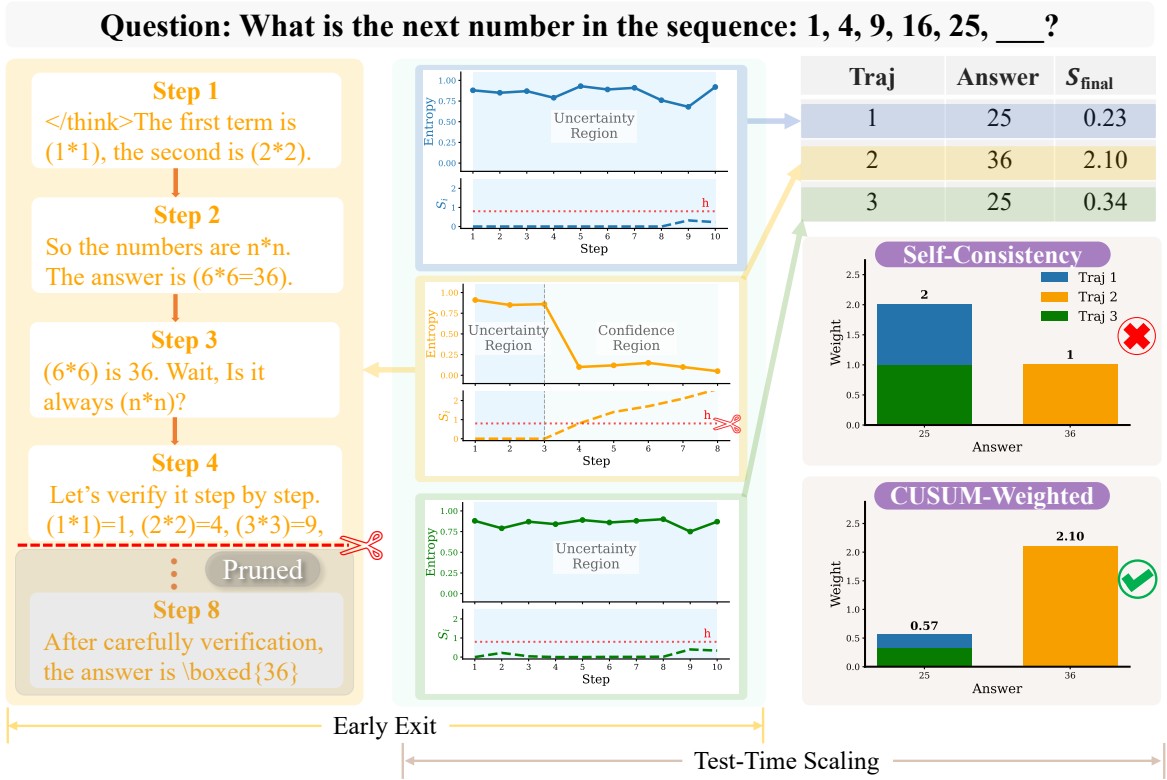

*Figure 2.* **Illustration of Confidence Region Detection for Efficient and Reliable Inference. (Center)** CUSUM statistics track the transition from the high-entropy Uncertainty Region to the low-entropy Confidence Region, where $S_i$ accumulates the log-likelihood ratio between the confidence and uncertainty hypotheses. **(Left) Early Exit:** By monitoring $S_i$ in real-time, we trigger early exit once cumulative evidence exceeds threshold $h$ (red dashed line). This enables pruning redundant computation immediately upon entering the Confidence Region while maintaining high reliability. **(Right) Test-Time Scaling:** The final CUSUM statistic $S_{\text{final}}$ serves as a trajectory quality indicator. Unlike Self-Consistency, which assigns equal weight to all trajectories, CUSUM-Weighted voting prioritizes trajectories that successfully converge to the Confidence Region (higher $S_{\text{final}}$ values), yielding more robust answer aggregation.

### 3.1. Optimization Objective

Our goal is to detect the Confidence Region transition as quickly as possible after it occurs (minimizing delay to exploit redundancy) while avoiding premature detection (controlling false alarms to preserve reliability).

Since the LLM generates tokens autoregressively, the detection rule must be online. We define the detection time $\tau$ as $\{\mathcal{F}_t\}$-adapted, where $\mathcal{F}_t = \sigma(\mathcal{H}_1, \ldots, \mathcal{H}_t)$ represents the information available up to step $t$. The optimization objective is:

$$\begin{aligned} \min_{\tau} \quad & \mathcal{D}_L(\tau) \\ \text{s.t.} \quad & \mathbb{E}_{\infty}[\tau] \geq \gamma, \end{aligned} \tag{4}$$

where $\mathcal{D}_L(\tau) \triangleq \sup_{\nu \geq 1} \operatorname{ess\,sup} \; \mathbb{E}_{\nu}[(\tau - \nu)^+ \mid \mathcal{F}_{\nu-1}]$ is the worst-case average detection delay (WADD); $\mathbb{E}_{\nu}[\cdot]$ and $\mathbb{E}_{\infty}[\cdot]$ denote expectations under a change at step $\nu$ and under no change, respectively; and $(\tau - \nu)^+ = \max(\tau - \nu, 0)$. The objective minimizes the worst-case detection

delay over all possible change-points $\nu$, while the constraint $\mathbb{E}_{\infty}[\tau] \geq \gamma$ controls false alarms by requiring the expected stopping time to be at least $\gamma$ when no change occurs (i.e., trajectories remain indefinitely in the Uncertainty Region).

This minimax formulation is motivated by two properties from Section 2:1. *Minimizing delay to exploit redundancy*: The Confidence Region's high redundancy (Figure 1) means detection delay wastes computation, so minimizing $\mathcal{D}_L(\tau)$ maximizes savings. 2. *Controlling false alarms to preserve reliability*: Given the sharp accuracy contrast ($< 20\%$ in Uncertainty vs. $> 60\%$ in Confidence, Figure 1), premature detection terminates reasoning while answers are unreliable. The constraint $\mathbb{E}_{\infty}[\tau] \geq \gamma$ bounds false alarms, ensuring safe termination.

### 3.2. Method: CUSUM

Following the two-regime model in Eq. (3) and the minimax objective in Eq. (4), we employ the Cumulative Sum (CUSUM) algorithm (Page, 1954), which is established as the asymptotically minimax-optimal solution. First, to

quantify evidence for the regime shift, CUSUM defines the per-step log-likelihood ratio:

$$Z_i = \log \frac{f_1(\mathcal{H}_i)}{f_0(\mathcal{H}_i)}, \tag{5}$$

which measures instantaneous evidence that the reasoning step $i$ belongs to the Confidence Region ($f_1$) rather than the Uncertainty Region ($f_0$).

**Definition 3.1** (CUSUM Statistic and Stopping Rule). The CUSUM statistic $S_i$ is defined recursively by:

$$S_0 = 0, \quad S_i = \max(0, S_{i-1} + Z_i), \quad i \geq 1. \tag{6}$$

Equivalently, $S_i = \max_{1 \leq k \leq i} \sum_{t=k}^{i} Z_t$, representing the maximum cumulative log-likelihood ratio at all possible starting points. We define the detection step using a pre-defined threshold $h > 0$, which controls the trade-off between detection speed and false-alarm robustness:

$$\tau_h = \inf\{i \geq 1 : S_i \geq h\}. \tag{7}$$

### 3.3. Feasibility: Identifiability Condition

For CUSUM to detect the Confidence Region, the two regions must be statistically distinguishable:

**Assumption 3.2** (Statistical Separability). The entropy distributions $f_0$ (Uncertainty Region) and $f_1$ (Confidence Region) satisfy:

$$\mathcal{D}_{\mathrm{KL}}(f_0 \| f_1) > 0 \quad \text{and} \quad \mathcal{D}_{\mathrm{KL}}(f_1 \| f_0) > 0 \tag{8}$$

where $\mathcal{D}_{\mathrm{KL}}(f \| g) = \int f(x) \log \frac{f(x)}{g(x)} dx$.

Necessity is immediate: indistinguishable distributions ($\mathcal{D}_{\mathrm{KL}}(f_0 \| f_1) = 0$) cannot be reliably detected by any algorithm. Sufficiency follows from the drift separation property of the log-likelihood ratio:

**Lemma 3.3** (Drift Separation). *Under Assumption 3.2, the log-likelihood ratio $Z_i = \log \frac{f_1(\mathcal{H}_i)}{f_0(\mathcal{H}_i)}$ has opposite expected signs:*

$$\begin{aligned} \mathbb{E}_0[Z_i] &= -\mathcal{D}_{\mathrm{KL}}(f_0 \| f_1) < 0, \\ \mathbb{E}_1[Z_i] &= \mathcal{D}_{\mathrm{KL}}(f_1 \| f_0) > 0. \end{aligned} \tag{9}$$

*Proof.* See Appendix A.2. □

This sign reversal enables CUSUM detection: $S_t$ drifts downward in the Uncertainty Region (staying below threshold) and upward in the Confidence Region (triggering detection at threshold $h$).

**Empirical Validation.** Table 4 confirms strictly positive KL divergences across all models, validating the feasibility of CUSUM-based detection.

---

**Algorithm 1** CUSUM-based Early Exit

---

**Input:** Input prompt $X$, Threshold $h$, Distribution estimates $f_0, f_1$
**Initialize:** $S_0 = 0$
**for** $i = 1, 2, \ldots$ **do**
    Generate next step $T_i$ and intermediate answer $A_i$
    Compute entropy $\mathcal{H}_i$, log-likelihood ratio $Z_i$
    Update CUSUM statistic: $S_i = \max(0, S_{i-1} + Z_i)$
    **if** $S_i \geq h$ **then**
        **break** {Detect Confidence Region Entry}
    **end if**
**end for**
**Return:** Generated trajectory $T_{1:i}$ and Answer $A_i$

---

**Algorithm 2** CUSUM-Weighted Test-Time Scaling

---

**Input:** Input prompt $X$, Number of trajectories $N$, Distribution estimates $f_0, f_1$
**Initialize:** Answer weights $W = \{\}$
**for** $n = 1$ **to** $N$ **do**
    Generate trajectory $\mathrm{Traj}^{(n)}$ of $L^{(n)}$ steps and extract final answer $A^{(n)}$
    Compute log-likelihood ratio sequence $\{Z_i^{(n)}\}$
    Compute final CUSUM score: $S_{\mathrm{final}}^{(n)} = \max_k \sum_{i=k}^{L^{(n)}} Z_i^{(n)}$
    Update weight for $A^{(n)}$: $W[A^{(n)}] \leftarrow W[A^{(n)}] + S_{\mathrm{final}}^{(n)}$
**end for**
**Return:** $\arg\max_a W[a]$

---

### 3.4. Properties: Theoretical Guarantees

CUSUM provides two key theoretical guarantees: (1) *controllable false alarm rate*—ensuring we don't terminate prematurely during exploration, and (2) *minimal detection delay*—stopping as soon as possible after convergence. Together, these properties maximize computational savings while maintaining reliability.

**Proposition 3.4** (False Alarm Control (Wald, 1947; Siegmund, 1985; Xu et al., 2021)). *Under the no-change measure $\nu = \infty$, the expected time to false alarm satisfies:*

$$e^h \leq \mathbb{E}_\infty[\tau_h] \leq C \cdot e^h \quad \text{as } h \to \infty, \tag{10}$$

*where $C$ is a constant depending on the KL divergence.*

This relationship enables precise false alarm control: setting $h = \log \gamma$ guarantees $\mathbb{E}_\infty[\tau_h] \geq \gamma$, preventing premature detection during the Uncertainty Region where answers are unreliable. Importantly, our method depends on only one hyperparameter $h$, which directly controls the reliability-efficiency trade-off. Increasing $h$ reduces false alarms (more reliable) but delays detection, while decreasing $h$ triggers earlier detection at the risk of premature false alarm. This simplicity prevents extensive hyperparameter search.

**Theorem 3.5** (Asymptotic Performance (Lorden, 1971)). *As $h \to \infty$, the CUSUM stopping rule $\tau_h$ achieves:*

$$\mathcal{D}_L(\tau_h) = \frac{\log \gamma}{\mathcal{D}_{\mathrm{KL}}(f_1 \| f_0)}(1 + o(1)). \quad (11)$$

This theorem suggests CUSUM approaches asymptotic optimality for confidence region detection. Among detection rules satisfying the false alarm constraint $\mathbb{E}_\infty[\tau] \geq \gamma$, CUSUM achieves near-minimal worst-case detection delay asymptotically. For CoT reasoning, this indicates our method can save close to the theoretical maximum computations from early exit.

The detection delay is inversely proportional to the KL divergence between regions: problems with sharper uncertainty-to-confidence transitions (higher $\mathcal{D}_{\mathrm{KL}}(f_1 \| f_0)$) tend to enable faster detection and greater savings. The linear dependence on threshold $h$ provides effective control over the reliability-efficiency trade-off, allowing practitioners to calibrate detection sensitivity based on computational budget and accuracy requirements.

**Robustness to Sequential Dependence.** The classical CUSUM framework assumes i.i.d. observations within each regime. In practice, the entropy sequence $\{\mathcal{H}_i\}$ exhibits temporal dependence across reasoning steps, as consecutive steps often share overlapping context. Fortunately, the consistency of CUSUM under dependent observations is well-established in the sequential analysis literature. Specifically, Kokoszka & Leipus (1998) show that CUSUM-type detectors remain consistent for identifying a mean shift under the mild condition that partial-sum variances grow sub-quadratically:

$$\mathrm{Var}\Big(\sum_{j=k}^{m} \mathcal{H}_j\Big) \leq C(m-k+1)^\delta, \quad 0 \leq \delta < 2. \quad (12)$$

In our setting, this condition is readily satisfied: the entropy drop at the regime shift is large relative to within-regime fluctuations (Table 4), yielding a high signal-to-noise ratio in the log-likelihood ratios $Z_i$ (Eq. 5). Consequently, the asymptotic guarantees established in Section 3.4 provide reliable approximations even in the presence of the sequential dependence inherent to CoT entropy sequences.

### 3.5. Implementations in Early Exit and Test-Time Scaling

To initialize the CUSUM detector, we first estimate the distributions $f_0$ and $f_1$ using Histogram Density Estimation (Freedman & Diaconis, 1981) on entropy samples from 100 randomly selected trajectories in the Bespoke-Stratos-17k dataset. Specifically, $f_0$ is fitted to entropies in the Uncertainty Region, and $f_1$ to those in the Confidence Region.

This calibration is **training-free**, requiring no model fine-tuning. Based on the CUSUM statistic $S_i$, we propose two algorithms for efficient inference:

**Early Exit.** As outlined in Algorithm 1 and Figure 2, we run CUSUM online along the CoT trajectory. The generation terminates as soon as $S_i \geq h$, signaling that the model has entered the Confidence Region with sufficient statistical evidence. This stopping rule $\tau_h$ leverages the minimax optimality of CUSUM to minimize token waste while ensuring reliability.

**Test-Time Scaling.** Algorithm 2 and Figure 2 detail our CUSUM-Weighted voting strategy. Instead of treating all sampled trajectories equally as in self-consistency (Wang et al., 2023), we score each trajectory $\mathrm{Traj}^{(n)}$ by its final CUSUM statistic $S_{\mathrm{final}}^{(n)}$. This score quantifies the cumulative evidence that the trajectory successfully converged to the Confidence Region. By weighting answers by $S_{\mathrm{final}}^{(n)}$, we prioritize high-quality, converged reasoning trajectories over those that remained in the Uncertainty Region.

## 4. Experiments

### 4.1. Experimental Setup

We evaluate three representative open-source LLMs of varying sizes: DeepSeek-R1-Distill-Qwen-7B, Qwen3-4B-Thinking-2507, and Qwen3-14B. We test these models on challenging reasoning benchmarks: AIME24 and AIME25 (competition-level mathematics) and GPQA-Diamond (Rein et al., 2023) (graduate-level science questions). All models use their standard prompting strategies, with full details in Appendix Figure 9. To ensure robust performance estimates, we vary the number of evaluation runs based on dataset size. For GPQA-Diamond, we average results over 4 random runs. For the smaller AIME datasets, we perform 16 random runs per dataset to mitigate sampling variance. Hyperparameter settings are detailed in Appendix B.1.

### 4.2. Early Exit

**Setup.** We compare CUSUM with three baselines: (1) Vanilla, which applies no early exit strategy and allows CoT reasoning to complete naturally; (2) DEER (Yang et al., 2025a), which employs a single-step high-confidence exit strategy; and (3) Dynasor (Fu et al., 2025a), which terminates generation upon detecting repetitive answer patterns.

**Results.** As shown in Table 2, CUSUM consistently outperforms both DEER and Dynasor across all three models and three datasets. On average, CUSUM achieves 63.06% accuracy with 12,413 tokens (11.1% reduction), outperforming DEER (59.78%) and Dynasor (58.70%) by 3.28 and 4.36

*Table 2.* Early Exit Results on AIME25, AIME24, and GPQA-Diamond across three models, showing accuracy (%) and token count with reduction percentage.

| Methods | AIME25 | | AIME24 | | GPQA | | Average | |
|---|---|---|---|---|---|---|---|---|
| | Acc ↑ | Tokens ↓ | Acc ↑ | Tokens ↓ | Acc ↑ | Tokens ↓ | Acc ↑ | Tokens ↓ |
| *DeepSeek-R1-Distill-Qwen-7B* | | | | | | | | |
| Vanilla | 41.04 | 14556 | 55.63 | 13313 | 38.38 | 8637 | 45.02 | 12169 |
| DEER | 37.5 | 12148↓16.5% | 46.67 | **10756**↓19.2% | 35.73 | 8227↓4.8% | 39.97 | 10377↓14.7% |
| Dynasor | 37.11 | 12822↓11.9% | 52.67 | 11215↓15.8% | 19.32 | 8342↓3.4% | 36.37 | 10793↓11.3% |
| Ours | **39.17** | **11740**↓19.3% | **53.75** | 10912↓18.0% | **40.40** | 8191↓5.2% | **44.44** | **10281**↓15.5% |
| *Qwen3-4B-Thinking-2507* | | | | | | | | |
| Vanilla | 81.04 | 22613 | 77.29 | 19178 | 64.65 | 9442 | 74.32 | 17078 |
| DEER | 76.2 | 21056↓6.9% | 76.0 | 18925↓1.3% | 64.65 | 9082↓3.8% | 72.28 | 16354↓4.2% |
| Dynasor | 74 | 20702↓8.4% | 75.56 | **18709**↓2.4% | 63.14 | **8968**↓5.0% | 70.9 | 16127↓5.6% |
| Ours | **78.13** | **20663**↓8.6% | **76.88** | 18714↓2.4% | **64.9** | 8980↓4.9% | **73.3** | **16119**↓5.6% |
| *Qwen3-14B* | | | | | | | | |
| Vanilla | 71.67 | 16727 | 81.46 | 13872 | 67.93 | 5988 | 73.69 | 12196 |
| DEER | 68.89 | 15438↓7.7% | 70.42 | 11454↓17.4% | 62.0 | 5942↓0.8% | 67.1 | 10944↓10.3% |
| Dynasor | 68.27 | **14305**↓14.5% | 76.22 | 12508↓9.8% | 61.99 | 5902↓1.4% | 68.83 | 10905↓10.6% |
| Ours | **68.96** | 14653↓12.4% | **77.71** | **12141**↓12.5% | **67.68** | **5720**↓4.5% | **71.45** | **10838**↓11.1% |

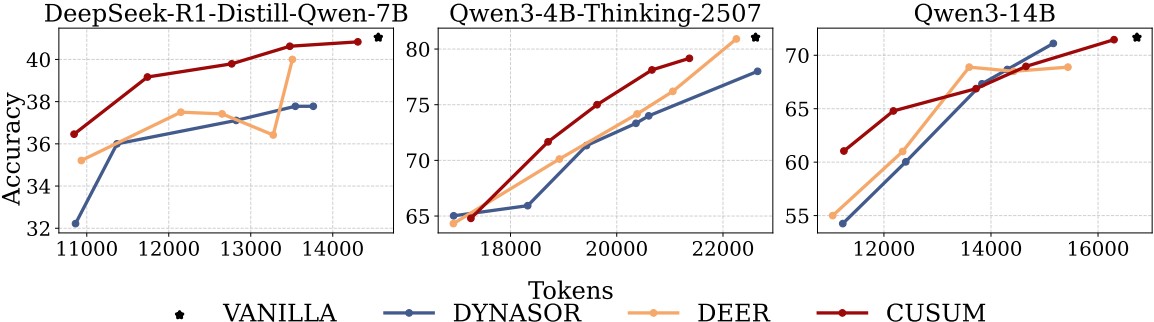

*Figure 3.* Pareto-frontier comparison of CUSUM, DEER, and Dynasor on AIME25: Our method achieves superior efficiency-accuracy trade-offs.

percentage points, respectively. The advantages are particularly pronounced in challenging scenarios: on GPQA with DeepSeek-R1-Distill-Qwen-7B, CUSUM achieves 40.40% accuracy, significantly outperforming DEER (35.73%) and Dynasor (19.32%).

While Table 2 reports performance at fixed configurations, a comprehensive evaluation requires examining the entire efficiency-accuracy trade-off space. To this end, we further compare the Pareto frontiers of different methods in AIME25. We vary the number of generated tokens by adjusting each method's key hyperparameter: the confidence threshold for DEER, the stable-answer threshold for Dynasor, and the threshold $h$ for CUSUM. The resulting Pareto frontiers, visualized in Figure 3, show that CUSUM achieves a superior Pareto frontier compared to the other methods. This consistent superiority across three diverse

models (4B to 14B parameters) demonstrates the robustness and generalizability of CUSUM.

We hypothesize that this improvement stems from CUSUM's principled detection of the uncertainty-to-confidence transition. While conventional heuristics often stop prematurely during uncertainty—sacrificing accuracy—or persist too long into redundancy, CUSUM adaptively identifies the confidence region. This allows for more aggressive pruning without compromising reliability.

**Computational Overhead.** To address the trade-off between probing latency and token reduction, we evaluate the total inference time of CUSUM, DEER, and Dynasor against a vanilla baseline on AIME25 using Qwen3-4B-Thinking-2507. As shown in Table 3, the overhead introduced by probing is remarkably minimal. These results

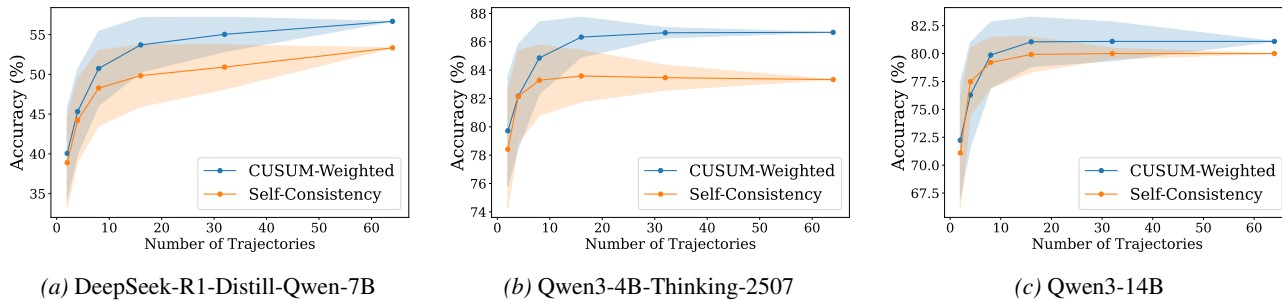

*Figure 4.* Test-time scaling results on AIME25 dataset comparing self-consistency across different numbers of sampled reasoning trajectories ($N = 2, 4, 8, 16, 32, 64$) for three models. CUSUM Weighted voting consistently outperforms self-consistency across three models, with the performance gap widening as the number of sampled trajectories increases. This highlights the effectiveness of leveraging CUSUM-based confidence metrics to prioritize high-quality reasoning trajectories.

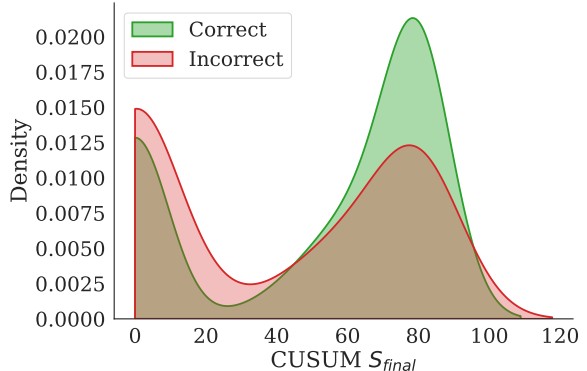

*Figure 5.* The distribution of CUSUM $S_{final}$ score from correct and incorrect samples.

*Table 3.* Computational Overhead of Different Early Exit Method.

| Method | Accuracy | Number of Tokens | Latency ($s$) |
|---|---|---|---|
| Vanilla | 41.04 | 14556 | 504 |
| DEER | 37.5 | 12148 | 460 |
| Dynasor | 37.11 | 12822 | 492 |
| Ours | 39.17 | 11740 | 419 |

demonstrate that the efficiency gains from reduced token usage significantly outweigh the marginal increase in latency, ensuring a net improvement in overall runtime.

### 4.3. Test-Time Scaling

**Setup.** For test-time scaling experiments, we generate $M = 64$ reasoning trajectories per problem using temperature sampling (temp = 0.8). We follow Lightman et al. (2024) to obtain the mean and variance for the Accuracy curve. Given a total of $M$ number of trajectories, for $N < M$, we calculate the mean and variance across all the samples of size $N$ from the $M$ trajectories. We compare against the **Self-Consistency** (Wang et al., 2023) baseline,

which uses standard majority voting where each trajectory receives equal weight. In contrast, our **CUSUM-Weighted** approach leverages the final CUSUM statistic $S_{final}$ as a quality indicator for each reasoning trajectory. We compute the weight score as in Algorithm 2.

**Results.** Figure 4 presents the results of our test-time scaling experiments across varying numbers of sampled reasoning trajectories ($N = 2, 4, 8, 16, 32, 64$) for all three models on AIME25. CUSUM-weighted voting consistently outperforms self-consistency across all models, demonstrating the robustness of our method.

The performance gap widens as the number of sampled trajectories ($N$) increases. Specifically, for Qwen3-4B-Thinking-2507, the improvement over self-consistency grows from a slim margin at $N = 2$ to a 3.33% lead at $N = 64$. This scaling behavior aligns with our insight: as more samples are available, the ability to prioritize trajectories that reached the Confidence Region over those stalled in the Uncertainty Region becomes increasingly critical.

**Validating $S_{final}$ as a Quality Indicator.** To verify that the final CUSUM statistic $S_{final}$ effectively captures trajectory quality, we analyze its distribution across correct and incorrect reasoning trajectories. Figure 5 shows the distributions for Qwen3-4B-Thinking-2507 on AIME25: correct trajectories exhibit higher $S_{final}$ values than incorrect ones. This clear separation demonstrates that $S_{final}$ serves as a reliable quality indicator—trajectories with strong evidence of entering the Confidence Region (high $S_{final}$) are more likely to be correct than those remaining in the Uncertainty Region (low $S_{final}$). Both correct and incorrect trajectories also exhibit a small peak near $S_{final} = 0$, which we attribute to persistent exploration without clear evidence accumulation for incorrect cases, and early truncation by the token limit for some correct cases.

These results underscore the value of entropy dynamics in test-time scaling. While standard Self-Consistency weighs

all trajectories equally, it often dilutes the correct signal with noise from "hallucinated" or unconverged trajectories. Our CUSUM score acts as a convergence filter, prioritizing trajectories that reach the Confidence Region. As sampling diversity increases, this ability to distinguish converged reasoning from exploration contributes largely to the performance gains, enabling more effective use of compute by focusing on higher-quality reasoning trajectories.

## 5. Related Work

### 5.1. Understanding Chain-of-Thought Reasoning

CoT reasoning has unlocked remarkable capabilities in Large Language Models (Xu et al., 2024; Shao et al., 2024; Bai et al., 2025; Wei et al., 2022; Xu et al., 2025b; Zheng et al., 2025; Hong et al., 2025). Recent work approaches understanding CoT through multiple lenses: Theoretically, Liu et al. (2024) prove that $T$ steps of CoT enable constant-depth transformers to simulate boolean circuits of size $T$, overcoming parallel processing constraints. Mechanistically, Dutta et al. (2024) identify a "functional rift" where representations transition from pre-training priors to in-context knowledge, while Li et al. (2025) introduce "CoT Vectors" revealing structured three-stage reasoning organization. Semantically, Yu et al. (2025) model reasoning as Markov chains to identify functional roles of individual steps. Most recently, Bachmann et al. (2026) estimate the potential of a CoT trajectory—the probability of eventually reaching the correct answer—via repeated sampling, providing an offline, post-hoc characterization of reasoning progress.

While these studies provide valuable insights into local properties and individual components, they either treat CoT as discrete segments or require computationally intensive offline analysis. Our work complements them by analyzing *online* global temporal dynamics—how reasoning evolves from exploration to convergence across the entire trajectory—using a single-pass entropy signal that enables real-time monitoring.

### 5.2. Early Exit for Chain-of-Thought

Training-free early-exit methods rely on heuristic signals to terminate CoT. Confidence-based approaches halt when answer confidence exceeds a threshold (Yang et al., 2025a), while stability-based methods stop once consecutive answers agree (Fu et al., 2025a). Others suppress exploration tokens when confident (Huang et al., 2025) or encourage brevity via prompts (Xu et al., 2025a). More recently, EAT (Wang et al., 2025b) detects convergence of next-token entropy near `</think>` via an EMA-variance heuristic, and HALT-CoT (Laaouach, 2025) applies a fixed entropy threshold at each step.

These methods improve efficiency but lack principled statistical foundations: per-step thresholding is vulnerable to transient low-entropy fluctuations in the Uncertainty Region (Section 2), and heuristic triggers offer no formal guarantees. Our work formulates early exit as sequential change-point detection over a predictive entropy signal that averages over intermediate answers, providing provable false-alarm control and minimax-optimal detection delay.

### 5.3. Test-Time Scaling for Chain-of-Thought

Test-time scaling leverages additional inference compute to improve reasoning performance (Snell et al., 2024). Self-consistency (Wang et al., 2023) aggregates multiple sampled trajectories via majority voting, while Brown et al. (2024) systematically compare strategies and find no single approach universally dominates. Concurrent works weight votes by final-answer confidence (Taubenfeld et al., 2025) or token-level confidence (Fu et al., 2025b).

Our approach instead derives trajectory quality from the entire convergence process: the CUSUM statistic captures *when* and *how decisively* a trajectory commits, providing a richer signal than point estimates of confidence for distinguishing converged from unconverged reasoning paths.

## 6. Conclusions

We present a systematic investigation of entropy dynamics in Chain-of-Thought reasoning, revealing a consistent two-phase structure: a sharp transition from an *Uncertainty Region* of stochastic exploration to a *Confidence Region* of deterministic convergence. The Confidence Region exhibits two critical properties—*High Reliability* and *High Redundancy*—that unlock efficient inference. Building on these insights, we are, to the best of our knowledge, the first to frame the monitoring of CoT reasoning as a classical sequential change-point detection problem. We adopt the CUSUM algorithm, yielding a training-free framework with minimax optimality guarantees. Our comprehensive evaluation across diverse models and benchmarks demonstrates practical value: in early-exit scenarios, we achieve superior Pareto frontiers with over 11% token reduction without compromising accuracy; in test-time scaling, CUSUM-weighted voting consistently outperforms self-consistency, with gains amplifying as sampling budget increases. This work provides an information-theoretic perspective on LLM reasoning dynamics, complementing existing analyses by focusing on global properties rather than local token-level features. By quantifying uncertainty reduction through entropy, we establish a principled approach for detecting reasoning convergence and identifying high-quality reasoning trajectories. These contributions advance practical methods for more efficient and reliable inference.

## Acknowledgement

The authors thank Bryan Wilie for his thoughtful comments and suggestions during the experiments and writing of this paper, which were helpful in improving its overall quality.

## Impact Statement

This research aims to enhance the computational efficiency and understanding of LLM reasoning. All experiments utilized publicly available, non-sensitive benchmark datasets (AIME24, AIME25, GPQA) and pre-trained models within their respective licenses.

We acknowledge the potential for dual-use concerns, as increased efficiency could inadvertently lower the barrier for malicious LLM applications, such as large-scale misinformation generation.

However, the primary impact of our work is to significantly reduce computational overhead, thereby promoting more environmentally sustainable AI and democratizing access to advanced LLM reasoning for researchers with limited resources. We are committed to responsible AI development and believe our contributions offer clear positive value to the community.

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

# A. Theoretical Proofs and Analysis

This appendix provides complete proofs and additional theoretical analysis for the results presented in the main text. We follow the notation established in Section 3.

## A.1. Notation Summary

We observe a sequence of entropy values $\mathcal{H}_1, \mathcal{H}_2, \ldots$ sequentially during CoT generation, where $\mathcal{H}_i \in [0, H_{\max}]$. There exists an unknown change-point $\nu \in \{1, 2, \ldots\} \cup \{\infty\}$ such that:

$$\mathcal{H}_i \sim \begin{cases} f_0, & i < \nu \quad \text{(Uncertainty Region)}, \\ f_1, & i \geq \nu \quad \text{(Confidence Region)}. \end{cases} \tag{13}$$

If $\nu = \infty$, no transition ever occurs and $\mathcal{H}_i \sim f_0$ for all $i$.

We use the following probability measures:

- $\mathbb{P}_\nu$: the measure under which change occurs at time $\nu$

- $\mathbb{P}_\infty$: the no-change measure, where all $\mathcal{H}_i \sim f_0$

- $\mathbb{P}_1$: the immediate-change measure where $\nu = 1$, so all $\mathcal{H}_i \sim f_1$

The log-likelihood ratio is $Z_i = \log \frac{f_1(\mathcal{H}_i)}{f_0(\mathcal{H}_i)}$, and the natural filtration $\mathcal{F}_i = \sigma(\mathcal{H}_1, \ldots, \mathcal{H}_i)$ represents the $\sigma$-algebra of events measurable with respect to observations up to time $i$.

## A.2. Proof of Identifiability (Lemma 3.3)

We prove that the Confidence Region is statistically detectable when $f_0$ and $f_1$ are separable (Assumption 3.2).

*Proof.* Let $f_0$ and $f_1$ denote the entropy distributions in the Uncertainty and Confidence Regions respectively. The log-likelihood ratio at step $t$ is defined as:

$$Z_i = \log \frac{f_1(\mathcal{H}_i)}{f_0(\mathcal{H}_i)} \tag{14}$$

**Expected Pre-change drift** ($\mathcal{H}_i \sim f_0$). When sampling from $f_0$, the expected log-likelihood ratio is:

$$\mathbb{E}_{f_0}[Z_i] = \int_0^{H_{\max}} \log \frac{f_1(h)}{f_0(h)} \cdot f_0(h) \, dh \tag{15}$$

$$= \int_0^{H_{\max}} f_0(h) \log f_1(h) \, dh - \int_0^{H_{\max}} f_0(h) \log f_0(h) \, dh \tag{16}$$

$$= -H(f_0, f_1) + H(f_0) \tag{17}$$

$$= -D_{KL}(f_0 \| f_1) < 0 \tag{18}$$

where $H(f_0, f_1)$ is the cross-entropy and $H(f_0)$ is the entropy of $f_0$. The inequality follows from the assumption $D_{KL}(f_0 \| f_1) > 0$ in Assumption 3.2.

**Expected Post-change drift** ($\mathcal{H}_i \sim f_1$). When sampling from $f_1$, the expected log-likelihood ratio is:

$$\mathbb{E}_{f_1}[Z_i] = \int_0^{H_{\max}} \log \frac{f_1(h)}{f_0(h)} \cdot f_1(h) \, dh \tag{19}$$

$$= \int_0^{H_{\max}} f_1(h) \log f_1(h) \, dh - \int_0^{H_{\max}} f_1(h) \log f_0(h) \, dh \tag{20}$$

$$= -H(f_1) + H(f_1, f_0) \tag{21}$$

$$= D_{KL}(f_1 \| f_0) > 0 \tag{22}$$

*Table 4.* KL-divergence between the entropy distributions of the uncertainty $f_0$ and confidence $f_1$ regions.

| Model | $D_{KL}(f_1\|f_0)$ | $D_{KL}(f_0\|f_1)$ |
|---|---|---|
| DeepSeek-R1-Distill-Qwen-7B | 0.87 | 1.42 |
| Qwen3-4B-Thinking-2507 | 1.31 | 2.78 |
| Qwen3-14B | 1.02 | 2.03 |

**Detectability guarantee.** The sign reversal in expected drift—negative before the change-point and positive after—is the fundamental requirement for sequential detection. Under these conditions:

- Before change ($i < \nu$): Negative drift keeps $S_i$ near zero, preventing false alarms.

- After change ($i \geq \nu$): Positive drift causes $S_i$ to accumulate linearly, eventually crossing threshold $h$.

This completes the proof that the regime shift is statistically detectable with finite expected delay. $\square$

### A.3. Statistical Properties of CoT Entropy Sequences

We provide additional statistical characterization of CoT entropy sequences that supports our theoretical framework.

**Bounded Support.** The predictive entropy $\mathcal{H}_i$ is naturally bounded: $\mathcal{H}_i \in [0, H_{\max}]$ where $H_{\max} = \log |V|$ depends on the vocabulary size $|V|$. In practice, for typical LLM vocabularies with $|V| \approx 150,000$, we have $H_{\max} \approx 11.9$ nats.

**Distribution Separation.** The KL-divergence between $f_0$ (Uncertainty Region) and $f_1$ (Confidence Region) quantifies the "signal strength" for detection. Larger divergence enables faster detection with lower error rates. Across our experiments:

- DeepSeek-R1-Distill-Qwen-7B: $D_{KL}(f_1\|f_0) \approx 0.87$ nats

- Qwen3-4B-Thinking-2507: $D_{KL}(f_1\|f_0) \approx 1.31$ nats

- Qwen3-14B: $D_{KL}(f_1\|f_0) \approx 1.02$ nats

## B. Experiments

### B.1. Hyperparameter Tuning

**Probing Interval $k$.** During CoT generation, we define a probing interval $k$ as the number of tokens generated in each decoding step $T_i$. We empirically evaluate intervals of 64, 128, and 256 tokens in early-exit setting. As shown in Table 5, an interval of 256 tokens leads to degraded accuracy, while an interval of 128 achieves comparable accuracy to 64 but requires fewer probing operations. Therefore, we select $k = 128$ as the optimal trade-off between efficiency and performance, balancing computational overhead with decoding reliability.

*Table 5.* Hyper-parameter tuning on the Probing Interval.

| Probing interval | 64 | 128 | 256 |
|---|---|---|---|
| Accuracy | 50.19 | 50.20 | 49.38 |
| #Tokens | 8,869 | 8,829 | 8,674 |

**Detection Threshold $h$.** Our early exit method mainly relies on the detection threshold $h$. We determine the optimal $h$ for each model using 100 randomly sampled instances from Bespoke-Stratos-17k as a validation set, prioritizing maximum token reduction while maintaining baseline accuracy.

Table 6 presents the selected thresholds. These values vary by model because the Kullback–Leibler (KL) divergence between the uncertainty and confidence distributions, $\mathcal{D}_{\mathrm{KL}}(f_0\|f_1)$, is model-specific (see Table 4).

*Table 6.* Hyper-parameter setting on $h$.

| Model | $h$ |
|---|---|
| DeepSeek-R1-Distill-Qwen-7B | 35 |
| Qwen3-4B-Thinking-2507 | 125 |
| Qwen3-14B | 65 |

The selected thresholds are higher than the theoretical value. According to Theorem 3.5, they correspond to theoretical delays of $\frac{h}{\mathcal{D}_{\mathrm{KL}}(f_1 \| f_0)} \approx$ 40-95 steps. This conservative choice is intentional: while CUSUM can detect the statistical transition into the Confidence Region rapidly, answer accuracy requires additional steps to fully stabilize. As shown in Figure 1, accuracy continues to improve for approximately 20 steps after the initial region transition. By setting higher $h$ values, we introduce a safety buffer ensuring termination occurs only after answers have converged to high reliability, rather than at the earliest statistical signal.

Importantly, we apply the same threshold $h$ across all datasets (AIME24, AIME25, GPQA-Diamond) for each model, without dataset-specific tuning. Despite significant variations in task difficulty—GPQA-Diamond focuses on graduate-level science questions while AIME targets high-school mathematics competitions—the fixed thresholds maintain consistent performance. This demonstrates that our method exhibits robustness across different reasoning domains and difficulty levels.

### B.2. Analysis on Different Difficulty Levels

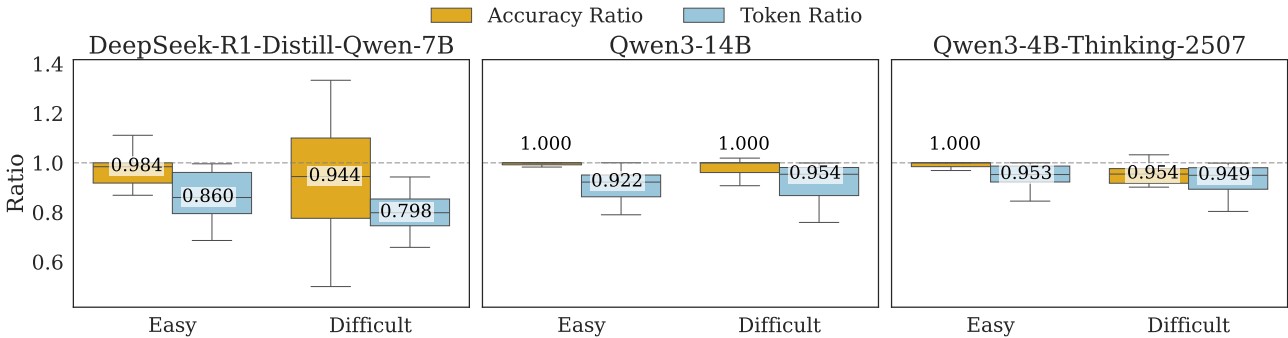

*Figure 6.* CUSUM efficiency across difficulty levels. Results show that CUSUM can reduce redundancy in easy problems without sacrificing accuracy across all models. For difficult instances, CUSUM achieves a dramatic 20.2% token saving on DeepSeek-R1-Distill-Qwen-7B, while the more optimized reasoning of Qwen3 models leads to more conservative exit behavior.

To analyze performance across problem difficulty, we partition AIME25 problems into "easy" and "difficult" subsets based on each model's vanilla accuracy. Figure 6 shows the accuracy ratio and token ratio relative to vanilla method. For easy problems, CUSUM achieves substantial token reduction with minimal accuracy loss across all models, demonstrating that models consistently generate redundant tokens even on problems they can solve reliably. For difficult problems, CUSUM's impact varies by model capability. DeepSeek-R1-Distill-Qwen-7B achieves dramatic token savings (20.2%) while maintaining comparable accuracy to vanilla. In contrast, the more powerful Qwen3 models show smaller token reductions, suggesting their vanilla reasoning on difficult problems already contains less redundancy and is more optimized.

### B.3. Trajectory Pattern Analysis

To understand how CUSUM handles different entropy dynamics, we classify each trajectory by the sign of its log-likelihood ratio sequence $\{Z_i\}$ into three categories:

- *Reliably transitioning* (36%): once $Z_i > 0$ is first observed, all subsequent $Z_j > 0$ for $j > i$.

- *Oscillating* (63%): after the first $Z_i > 0$, at least one subsequent $Z_j < 0$, meaning the trajectory re-enters the Uncertainty Region before final convergence.

- *Never converging* (1%): $Z_i < 0$ throughout the entire generation.

*Table 7.* Stratified comparison by trajectory pattern on AIME25 (DeepSeek-R1-Distill-Qwen-7B).

| Method | Reliably Trans. (36%) | | Oscillating (63%) | | Never Conv. (1%) | |
|---|---|---|---|---|---|---|
| | Acc | Tokens | Acc | Tokens | Acc | Tokens |
| Vanilla | 74.6 | 6518 | 55.6 | 11027 | 0 | 32678 |
| DEER | 71.34 | 6132 | 41.0 | 9120 | 0 | 32678 |
| Dynasor | 72.42 | 5878 | 46.93 | 9029 | 0 | 32678 |
| **CUSUM** | **74.6** | **6239** | **54.1** | **9375** | 0 | 32678 |

This classification is deterministic and algorithm-agnostic. Table 7 reports the stratified comparison on AIME25 with DeepSeek-R1-Distill-Qwen-7B.

For reliably transitioning trajectories, all methods reduce tokens since the entropy signal is unambiguous. However, CUSUM is the only method that preserves full Vanilla accuracy (74.6%) while achieving 4.3% token reduction, as it waits for statistically confirmed convergence rather than exiting at the first low-entropy observation.

For the oscillating subset, which constitutes the majority (63%) of trajectories, the performance gap becomes substantial. Threshold-based methods (DEER, EAT, HALT-CoT) exit prematurely at transient low-entropy dips, losing 9–14 percentage points in accuracy. Dynasor is more robust due to its stable-answer criterion, but still loses 8.7 points. In contrast, CUSUM loses only 1.5 points because $S_i$ resets toward zero during high-entropy phases and accumulates only under sustained low-entropy evidence. This demonstrates that cumulative evidence accumulation is essential for distinguishing genuine convergence from transient fluctuations.

For the rare never-converging cases (1%), no method triggers an exit, and all degrade to Vanilla generation with 0% accuracy. This confirms that CUSUM degrades safely when no convergence occurs.

## B.4. Analysis: Correct vs. Incorrect Convergence

To understand CUSUM's behavior when the model enters the Confidence Region, we analyze trajectories based on their final answer correctness. We categorize trajectories into two types: *correct convergence* (entering the Confidence Region with correct answers) and *incorrect convergence* (entering the Confidence Region with wrong answers). Table 8 compares vanilla generation and CUSUM early exit for DeepSeek-R1-Distill-Qwen-7B on AIME25.

*Table 8.* Analysis of trajectories that enter the Confidence Region: accuracy preservation and token consumption for DeepSeek-R1-Distill-Qwen-7B on AIME25.

| Method | Correct Convergence (Acc% / Avg. Tokens) | Incorrect Convergence (Acc% / Avg. Tokens) |
|---|---|---|
| Vanilla (Full Generation) | 100.0 / 7,516 | 0.0 / 19,210 |
| CUSUM Early Exit | 95.71 / 6,541 | 0.0 / 15,285 |
| *Token Reduction* | *13.0%* | *20.4%* |

**Correct Convergence.** For trajectories that converge to correct answers, CUSUM early exit preserves 95.71% accuracy while reducing token consumption by 13.0% (from 7,516 to 6,541 tokens). This demonstrates the core value proposition of our method: by detecting the transition to the Confidence Region, we can safely terminate generation once the model has converged to the correct answer, eliminating redundant computation without sacrificing correctness.

**Incorrect Convergence.** For trajectories that converge to incorrect answers, CUSUM early exit maintains 0% accuracy (as expected—the model has already settled on the wrong answer) while still reducing token consumption by 20.4% (from 19,210 to 15,285 tokens). This shows that CUSUM successfully identifies convergence regardless of answer correctness, reducing latency even on failure cases without introducing additional errors.

**Key Insight.** This analysis reveals a critical property: CUSUM detects the *convergence event* itself, independent of answer correctness. Once the model enters the Confidence Region, continued generation yields minimal benefit—whether

*Table 9.* Generalization to Open-Ended Tasks with Qwen3-4B-Thinking-2507. The same threshold $h$ calibrated on Bespoke-Stratos-17k is applied without recalibration.

| Method | HotpotQA | | LiveCodeBench | |
|---|---|---|---|---|
| | EM ↑ | Tokens ↓ | ACC ↑ | Tokens ↓ |
| Vanilla | 34.95 | 3511 | 90.92 | 2898 |
| DEER | 33.20 | 3392 | 86.60 | 2683 |
| Dynasor | 23.20 | 2378 | 85.90 | 2596 |
| **CUSUM** | **34.10** | **3080** | **86.60** | **2621** |

the answer is correct or incorrect, the reasoning has stabilized. By exploiting this high redundancy property (Section 2), CUSUM achieves consistent efficiency.

### B.5. Generalization to Open-Ended Tasks.

To evaluate whether CUSUM generalizes across domains, we apply it to two Open-Ended benchmarks using Qwen3-4B-Thinking-2507: HotpotQA (Yang et al., 2018) (multi-hop Wikipedia QA with free-form entity spans) and LiveCodeBench (Jain et al., 2025) (code execution output prediction with Python literals). Both benchmarks differ from the calibration set (Bespoke-Stratos-17k) in task type and answer format.

As shown in Table 9, CUSUM achieves a consistently superior accuracy-efficiency trade-off. On HotpotQA, CUSUM reduces tokens by 12.3% while incurring only a 0.85 EM drop relative to Vanilla; in contrast, DEER and Dynasor sacrifice 1.75 and 11.75 EM respectively. On LiveCodeBench, CUSUM matches DEER in accuracy (86.6%) while using 2.3% fewer tokens.

### B.6. Performance on Short Reasoning Chains

To assess CUSUM's effectiveness on simpler problems with short reasoning chains, we evaluate it on the AMC23 dataset, which contains elementary mathematics questions requiring shorter reasoning chains. Table 10 shows results for DeepSeek-R1-Distill-Qwen-7B.

*Table 10.* Performance on AMC23 dataset with short reasoning chains using DeepSeek-R1-Distill-Qwen-7B.

| Method | Accuracy (%) | Avg. Tokens |
|---|---|---|
| Vanilla | 89.69 | 6,233 |
| CUSUM | 89.36 | 5,629 |
| *Change* | *-0.33* | *-9.7%* |

Even on short reasoning chains, CUSUM achieves comparable accuracy (89.36% vs. 89.69%) while reducing token consumption by 9.7%. This demonstrates that the Confidence Region detection mechanism generalizes beyond complex multi-step reasoning: models generate redundant tokens even on simpler problems, and CUSUM successfully identifies when reasoning has converged regardless of chain length.

## C. Author Contributions

Ting Xu led the project, designed and conducted all experiments, and wrote the manuscript. Xu He co-designed the research framework and provided technical guidance. Yupu Lu developed the theoretical analysis including optimality guarantees in Section 3. Jiankai Sun provided feedback on method development and technical guidance. Dong Li, Wai Lam, and Jianye Hao offered research direction and feedback on experimental design. All authors reviewed and approved the final manuscript.

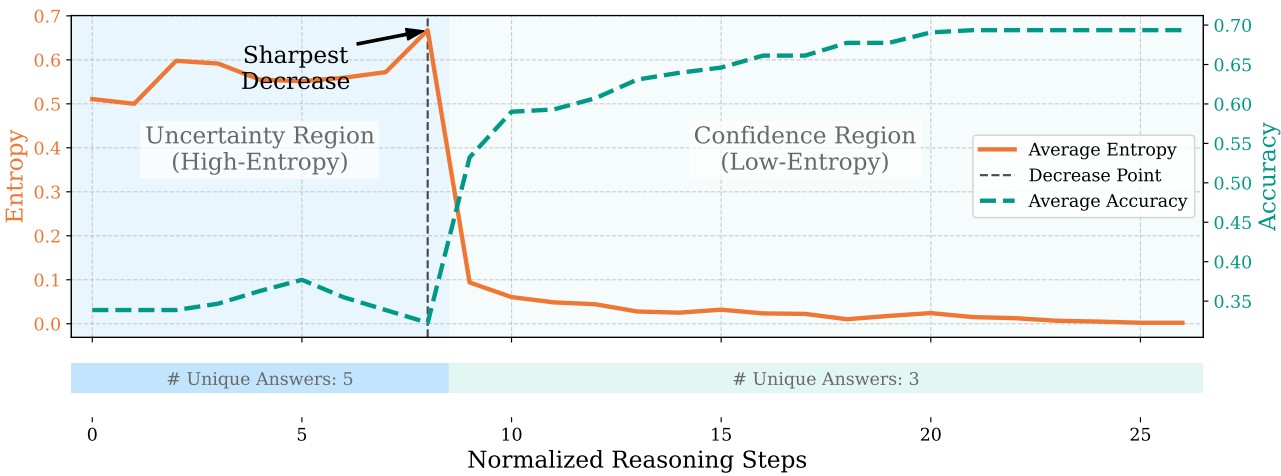

*Figure 7.* Entropy and Accuracy dynamics across reasoning steps for DeepSeek-R1-Distill-Qwen-7B.

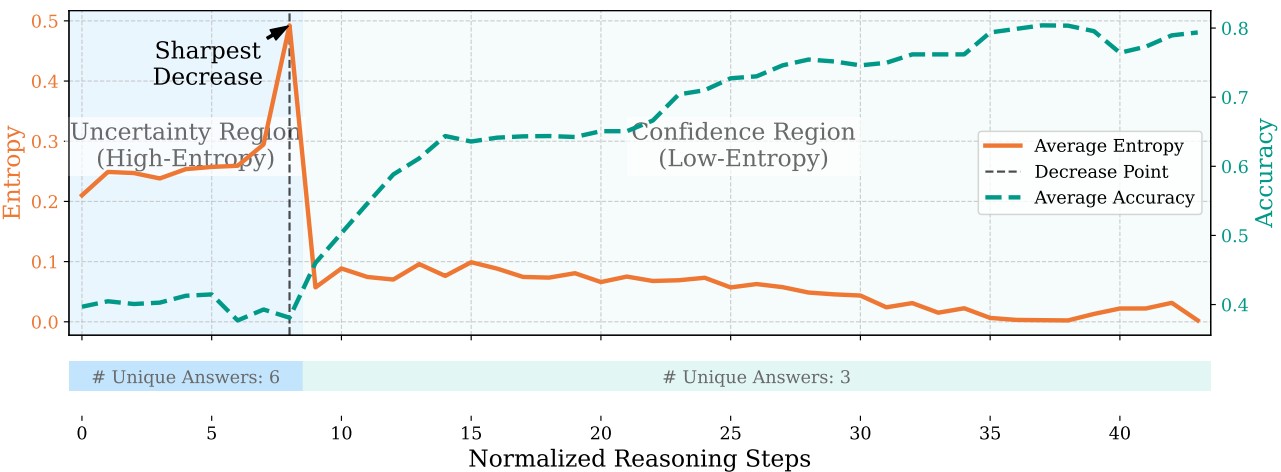

*Figure 8.* Entropy and Accuracy dynamics across reasoning steps for Qwen3-14B.

## D. The Use of LLMs

In the preparation of this manuscript, LLMs were utilized as a writing assistant to enhance clarity, refine phrasing, and improve overall readability. Specifically, LLM tools were employed for:

- Grammar and Style Refinement: Identifying and correcting grammatical errors, improving sentence structure, and suggesting more concise or academic phrasing.

- Vocabulary Enhancement: Proposing alternative word choices to avoid repetition and enrich the lexical diversity of the text.

- Conciseness: Restructuring sentences or paragraphs to ensure that arguments are presented in a clear and succinct manner.

It is important to note that LLMs were used solely for **editorial support and linguistic enhancement**. All research ideas, experimental design, data analysis, interpretation of results, and the core scientific content presented in this paper are the original work of the authors. The LLM's role was strictly limited to improving the articulation of these original contributions, not to generate any scientific content or insights. The authors have meticulously reviewed and approved all LLM-assisted revisions to ensure accuracy and alignment with their intended meaning.

**Prompt for AIME24 and AIME25**

<|im_start|>system
Please reason step by step, and put your final answer within \\boxed{}.
<|im_end|>
<|im_start|>user
{{content}}
<|im_end|>
<|im_start|>assistant

- - - - - - - - - - - - - - - - - - - - - - - - - - - - - - - - - - - - - -

**Prompt for GPQA Diamond**

<|im_start|>system
Return your final response within \\boxed{} and only include the letter choice (A, B, C, or D) as your final response.
<|im_end|>
<|im_start|>user
{{content}}
<|im_end|>
<|im_start|>assistant

*Figure 9.* Prompts for different dataset.

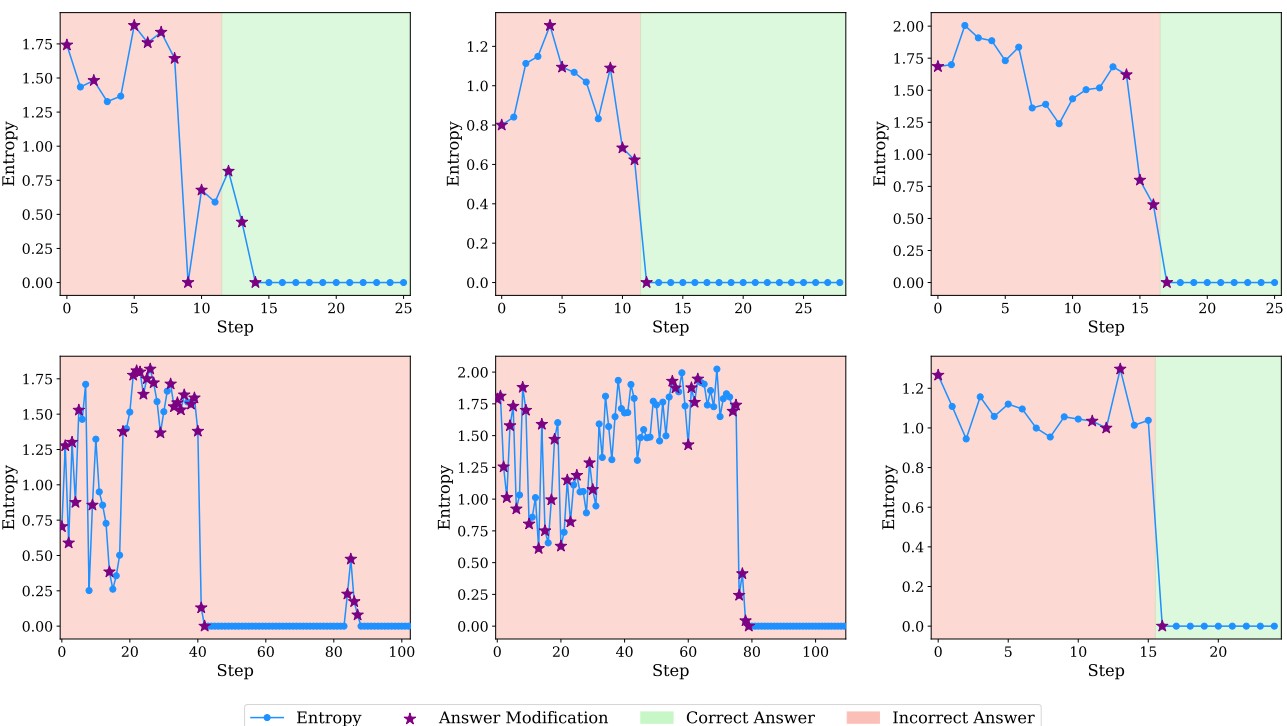

*Figure 10.* An illustration of CoT dynamics across six cases. In each subplot, the blue line tracks the model's entropy (y-axis) at each reasoning step (x-axis). **Background shading** indicates the correctness of the intermediate answer when compared to the ground truth (green for correct, red for incorrect). **Purple stars** ★ mark the exact steps where the model modifies its answer. We consistently observe a pattern of the entropy: (1) an initial high-entropy plateau, (2) a sharp entropy decrease, and (3) a stable low-entropy plateau. This suggests that CoT reasoning bifurcates abruptly rather than evolving gradually from exploration to convergence.

# E. Reproducibility Statement

To facilitate the reproducibility of our findings, we provide a detailed account of our methodology, implementation, and experimental setup. The core logic and formulation of our proposed method, CUSUM, are described in Section 3. For implementation-specific details, the exact prompts used for different dataset is in Figure 9 in the Appendix. A comprehensive list of all hyperparameters, including the probing interval ($k$) and CUSUM threshold $h$ used for each model, is provided in Table 5 and 6. All datasets used in our evaluation (AIME25, AIME24, and GPQA) are publicly available. To ensure the stability and robustness of our findings, all experiments were repeated multiple times, and the averaged results are reported. Our source code will be made publicly available upon acceptance of this manuscript.

