# OpenReview forum: "Unveiling the Entropy Dynamics of Chain-of-Thought Reasoning"
_ICML.cc/2026/Conference — ICML 2026 regular_

### Official Review · Reviewer_TZAN · 2026-03-06

**Soundness:** 3
**Presentation:** 3
**Significance:** 2
**Originality:** 4
**Overall Recommendation:** 3
**Confidence:** 5

**Summary:**

This paper investigates the dynamics of the sequence of output tokens for a given prompt in LLMs. The "chain of thought" is a stochastic process and the empirical conditional entropy also forms a stochastic process. The authors demonstrate that the latter sequence changes from a large value to a smaller value in a phase-transition like phenomenon. The paper investigates the change point, using the variants of the CUSUM algorithm. They provide a method that achieves quickest detection characteristics.

**Compliance With Llm Reviewing Policy:**

Affirmed.

**Key Questions For Authors:**

None.

**Limitations:**

Yes

**Strengths And Weaknesses:**

Strengths:
1. The problem is novel and interesting; and the phase transition observation is nice.
2. The predictive entropy metric used seems original.
3. Presentation is clear throughout the paper.

Weaknesses:
1. The detection method is rather standard.
2. The performance theorems and lemmas are not novel (can be merely cited) and at times quite straightforward (Lemma 3.3 is obvious and does not need a proof).
3. In my opinion, the core problem and the dynamics of the empirical conditional entropy in LLMs is much more interesting than the change detection problem. It would make the problem much more original if the authors dissected that dynamics and provide more insights into the phase transition phenomenon rather than the rather straightforward details of how to quickly detect that point.
4. Not a major weakness, but it would be good to use more standard notation for predictive entropy metric. The notation makes it look like a conditional entropy, but it actually is some sort of measured empirical entropy.

---

> ### Author Rebuttal · Authors · 2026-03-31
>
> Dear Reviewer TZAN,
>
> Thank you for the comment, we will address your concerns below.
>
> **Q1: Standard detection method and non-novel theorems.**
>
> A1: We agree that CUSUM (Page, 1954; Lorden, 1971) is a classical algorithm and that the theoretical results in Section 3 are established results from sequential analysis. Our novelty claims rest on four contributions that are independent of CUSUM itself: (1) the **empirical discovery** of the two-phase structure in CoT; (2) the **formal modeling** of this structure as a sequential change-point problem, connecting CoT analysis to sequential analysis theory for the first time; (3) the **design of the predictive entropy signal** (Eq. 2) as the observable for CUSUM; and (4) **practical algorithms** for early exit and test-time scaling with experimental validation across diverse models and benchmarks.
>
> The theorems are presented not as original mathematical contributions but to explicitly verify that classical guarantees carry over to our specific setting, and to interpret their meaning for CoT monitoring (false-alarm control → no premature exit; minimax optimality → maximal token savings). All proofs remain in the Appendix; the main-text theorems focus entirely on their interpretation. We will revise the paper to make these novelty claims sharper.
>
> **Q2 Provide more insights into the phase transition phenomenon.**
>
> A2: We thank the reviewer for this suggestion—it aligns with our own view that the phenomenology is the most interesting part. We will expand Section 2 with the following three new analyses:
>
> | Model | Transition point $\nu$ (easy) | Transition point $\nu$ (hard) | Corr($\nu$, ACC) |
> |---|---|---|---|
> | DeepSeek-R1-Distill-Qwen-7B | 33.9% | 56.4% | −0.45 |
> | Qwen3-4B-Thinking-2507 | 31% | 54% | −0.36 |
> | Qwen3-14B | 30% | 52% | −0.26 |
>
> *Transition point expressed as normalized position along the reasoning chain.*
>
> **[Finding 1] Problem difficulty governs when the model commits.** On easy problems, the phase transition occurs in the first third of the reasoning chain (30–34%); on hard problems it is delayed to the middle-to-late stage (52–56%). This reveals that *redundancy is predominantly a phenomenon of easy problems*: once the model commits early, it continues generating well past convergence. This has a direct practical implication: harder problems yield less redundancy to exploit—consistent with our results in Figure 6.
>
> **[Finding 2] Stronger models commit earlier and more decisively.** Across all difficulty levels, stronger models exhibit earlier transition points (Qwen3-14B: 30%/52% vs. DeepSeek-R1-7B: 33.9%/56.4%). This suggests that model capacity directly accelerates the exploration-to-convergence transition — stronger models need fewer reasoning steps to resolve uncertainty.
>
> **[Finding 3] Earlier convergence predicts correctness, especially for distilled models.** The negative correlation between transition point ν and final accuracy (Corr = −0.45/−0.36/−0.26 for 7B/4B/14B respectively) reveals that *when* a model commits is a strong signal of *whether* it will be correct. Strikingly, this correlation is strongest for the distilled 7B model and weakest for the natively trained 14B model. We interpret this as a signature of knowledge distillation: distilled models have less robust internal representations, so their convergence timing is more tightly coupled to whether their reasoning is actually correct. Native reasoning models (Qwen3-14B) can arrive at correct answers through longer, more exploratory paths, making their transition point a weaker correctness predictor. This has a direct implication for early exit: the CUSUM threshold $h$ should arguably be set more conservatively for distilled models, which our empirical tuning (h=35 for 7B vs. h=65 for 14B, Table 4) independently confirms.
>
> We will incorporate these analyses into an expanded Section 2 in the revision.
>
> **Q3 Not a major weakness, but it would be good to use more standard notation for predictive entropy metric.**
>
> Thank you for this suggestion. We will rename this quantity $\hat{H}_i$ (empirical predictive entropy) and add a clarifying note at its first definition (Eq. 2) distinguishing it from conditional entropy.

---

> > ### Author Rebuttal · Reviewer_TZAN · 2026-04-01
> >
> > I like the general problem space and the specific application the authors have identified - dynamics of empirical conditional entropy in LLM space. However, I do not find the main focus of change point detection problem and its application as a novel technical contribution. So, formal modeling of the structure as a change-point detection problem and the design of an observable for CUSUM are not as major in my opinion. The paper has a lot of merit if it studies the fundemental insights for empirical conditional entropies (not certain numerical evaluations for specific models). That could be done, but in a different paper in my opinion.
> >
> > I will still keep my Weak Reject recommendation, with a strong encouragement to the authors to add on to the foundations in a somewhat different direction as specified above.

---

> > > ### Author Response · Authors · 2026-04-08
> > >
> > > Dear Reviewer TZAN,
> > >
> > > Thank you for the feedback. While we respect your perspective on the technical contribution, we wish to clarify how our framework bridges the gap between qualitative phenomenology and principled engineering:
> > >
> > > 1. From Observation to Structural Insight: Our new analyses (Findings 1-3) move exactly in the foundational direction you envisioned. For example, the discovery that distilled models show a tighter coupling between convergence timing and accuracy (Corr = −0.45) than native models (−0.26) reveals that the transition point $\nu$ is a structural signature of internal representations. This provides deep insight into CoT dynamics.
> > >
> > > 2. Conceptual Innovation: Although CUSUM is classical, its application here is not "standard." Our Predictive Entropy (Eq. 2) is a novel observable specifically designed to map the high-dimensional, stochastic CoT process into a 1D signal suitable for formal analysis.This mapping is what allows us to move from a heuristic, "trial-and-error" observation to a predictive framework with minimax-optimal guarantees.
> > >
> > > 3. Controllable Systems: Without a formal detector, the two-phase structure remains a passive observation. Our change-point formulation converts this into a controllable property, enabling explicit, minimax-optimal trade-offs between false-alarms (premature exit) and detection delay.
> > >
> > > We believe this transition—from "heuristic observation" to "controllable system"—is a substantive step toward rigorous LLM engineering. We are grateful for your Originality: 4 assessment and for encouraging this more foundational treatment.

---

### Official Review · Reviewer_q8Ci · 2026-03-13

**Soundness:** 3
**Presentation:** 3
**Significance:** 3
**Originality:** 2
**Overall Recommendation:** 4
**Confidence:** 3

**Summary:**

This paper studies the overthinking problem in reasoning LLMs. The authors observe that when intermediate reasoning traces are truncated and the model is prompted to answer, the predictive entropy often exhibits a sharp transition from a high-entropy uncertainty region to a low-entropy confidence region. Based on this observation, the paper formulates confidence-region detection as a sequential change-point detection problem and applies a CUSUM-based stopping rule for early exit. The paper also explores a confidence-aware test-time scaling strategy based on the final CUSUM statistic.

**Compliance With Llm Reviewing Policy:**

Affirmed.

**Final Justification:**

I would like to thank the authors for the substantial additional experiments in the rebuttal and for providing further results on oscillating trajectories. The authors have effectively addressed my concerns. In particular, the experimental results on oscillating trajectories, including comparisons with the baselines, make CUSUM’s algorithmic advantage clearer.

Based on these additional experiments and clarifications, I am raising my score to weak accept.

**Key Questions For Authors:**

1. Does predictive entropy reliably transition from a high state to a low state? Are there specific cases or tasks where the confidence metric oscillates, or fails to reach a high level before the generation length limit is reached?
1. Could you provide detailed information on how hyperparameter tuning was conducted for the baseline methods? If it was not conducted, how do you justify the current performance comparisons?
1. How does the proposed CUSUM-based change-point detection specifically outperform the entropy-based early stopping mechanisms proposed in [1] and [2]?

**Limitations:**

yes

**Strengths And Weaknesses:**

## Strengths
1. The computational cost associated with long reasoning in reasoning LLMs is a well-known and practically important problem.
1. The core observation that reasoning can often be truncated with limited loss once the model appears to have converged is practically important.
1. The authors empirically demonstrate that their proposed method achieves a superior Pareto frontier on AIME25 across three model sizes.
## Weaknesses
1. **Fairness of empirical comparison**: The proposed method uses model-specific hyperparameters tuned on a validation set, but the paper does not sufficiently document how the baseline methods were tuned. For a rigorous comparison, the tuning protocol for all methods should be described clearly and made as comparable as possible.
1. **Limited novelty relative to recent works**: The paper does not sufficiently discuss several highly relevant recent works on entropy/confidence-based early stopping and confidence-aware trajectory aggregation. In particular, [1, 2] appear closely related to the early-exit component, while [3, 4] suggest that confidence-aware aggregation for test-time scaling is already an active area. The paper should more clearly articulate what is genuinely new about the CUSUM formulation beyond these prior approaches.

[1] EAT: Entropy After /think for reasoning model early exiting https://arxiv.org/abs/2509.26522

[2] HALT-CoT: Model-Agnostic Early Stopping for Chain-of-Thought Reasoning via Answer Entropy https://openreview.net/forum?id=CX5c7C1CZa

[3] Confidence Improves Self-Consistency in LLMs https://aclanthology.org/2025.findings-acl.1030/

[4] Deep Think with Confidence https://arxiv.org/abs/2508.15260

---

> ### Author Rebuttal · Authors · 2026-03-31
>
> Dear Reviewer q8Ci,
>
> Thank you for the detailed feedback, we will address your concerns below.
>
> **Q1 Fairness of empirical comparison and hyperparameter tuning protocol.**
>
> A1: For CUSUM, threshold $h$ is tuned on 100 instances from Bespoke-Stratos-17k (not the test benchmarks), selecting from $h \in \\{20, 35, 45, 65, 85, 105, 125\\}$
> to maximize token reduction while maintaining accuracy comparable to vanilla (Appendix B.1).
>
> For DEER and Dynasor, we followed the recommended hyperparameter settings from their respective papers, then swept their key hyperparameters across the equivalent Pareto range for fair comparison:
> - DEER's confidence threshold $\lambda=0.95$, then swept across [0.5, 0.75, 0.9, 0.95, 0.99] to cover a range of accuracy-efficiency trade-offs.
> - Dynasor's stable-answer threshold $\mathcal{H}_T = 0.7$, then swept over [0.4, 0.5, 0.6, 0.7, 0.8] to cover a similar range.
>
> Crucially, Figure 3 compares full Pareto frontiers rather than single operating points, so no method is disadvantaged by hyperparameter
> choice. CUSUM achieves a superior Pareto frontier.
>
> **Q2 Limited novelty relative to early exit methods [1, 2], and confidence-aware aggregation [3,4]**
>
> A2: CUSUM's advantage is grounded in statistical optimality theory that prior approaches lack. Our two-phase structure observation also simultaneously unlocks both early exit and test-time scaling within a unified framework — whereas [1–4] each address only one application in isolation.
> - vs. EAT [1]: EAT identifies a monotonic entropy decrease near `</think>`. Our work complements this by characterizing a **two-phase structure** throughout the reasoning chain, which motivates a formal change-point model. On the signal side, EAT monitors next-token entropy, while our predictive entropy (Eq. 2) averages over intermediate answers — a design choice aimed at reducing sensitivity to local token-level fluctuations.  Theoretically, EAT uses a heuristic EMA-variance threshold with no guarantees, while CUSUM provides minimax-optimal detection (Theorem 3.5) and provable false-alarm control (Proposition 3.4).
> - vs. HALT-CoT [2]: HALT-CoT applies a fixed entropy threshold independently at each step, making it sensitive to transient low-entropy fluctuations in the Uncertainty Region where answer accuracy is low (Figure 1). CUSUM accumulates evidence across steps, providing robustness to these false signals and formal statistical guarantees.
> - vs. confidence-aware aggregation [3,4]:  [3] focuses on the confidence of the final answer, while our method tracks cumulative evidence of a distributional shift throughout the trajectory. [4] operates at the token level; our answer-level probing is a deliberate design choice intended to reduce noise, though we acknowledge both levels of granularity have merit.
>
> We compare [1] and [2] on AIME25 with DeepSeek-R1-Distill-Qwen-7B:
> ||Acc|Tokens|
> |---|---|---|
> |Vanilla|41.04|14556|
> |Ours|39.17|11740|
> |EAT|35.21|9280|
> |HALT-CoT|26.67|10060|
>
> CUSUM achieves the highest accuracy among early-exit methods, outperforming EAT by +3.96 pp and HALT-CoT by +12.50 pp. HALT-CoT's −14.37 pp accuracy drop directly illustrates the premature-exit failure mode that CUSUM's evidence accumulation prevents. Note that Vanilla remains the accuracy ceiling; CUSUM recovers 97.9% of Vanilla accuracy while reducing tokens by 19.3%.
>
> [1] EAT: Entropy After /think for reasoning model early exiting
>
> [2] HALT-CoT: Model-Agnostic Early Stopping for Chain-of-Thought Reasoning via Answer Entropy
>
> [3] Confidence Improves Self-Consistency in LLMs
>
> [4] Deep Think with Confidence
>
> **Q3 Does entropy reliably transition high-to-low, or does it oscillate / fail to converge within the generation limit?**
>
> A3: We categorize all trajectories into three behavioral patterns based on their entropy dynamics:
>
> |Pattern | Reliably high→low | Oscillating  high → low | Never converging |
> |---|---|---|---|
> | Proportion | 36% | 63% | 1% |
> | Vanilla (Acc/Tokens) | 74.6 / 6518 | 55.6 / 11027 | 0 / 32678 |
> | Ours (Acc/Tokens) | 74.6 / 6239 | 54.1 / 9375 | 0 / 32678 |
>
>
> **Oscillating (63%)**: Unlike simple thresholding — which would exit on any transient low-entropy points — CUSUM's statistic $S_i$​ resets toward zero during high-entropy phases and accumulates only under sustained low-entropy evidence. CUSUM achieves 54.1% accuracy on these trajectories, closely matching vanilla (55.6%), confirming that the method does not prematurely exit during oscillation.
>
> **Reliably transitioning (36%)**:  Full accuracy preserved (74.6% vs. 74.6%) with 4.3% token reduction, confirming safe termination upon genuine convergence.
>
> **Never converging (1%)**: CUSUM correctly never fires ($\nu=\infty$), falling back to vanilla generation with no accuracy penalty.
>
> In summary, CUSUM handles all three trajectory types gracefully: it saves compute on reliably transitioning trajectories, avoids false exits during oscillations, and degrades safely to vanilla on non-converging cases.

---

> > ### Author Rebuttal · Reviewer_q8Ci · 2026-04-04
> >
> > Thank you for the detailed rebuttal and for addressing my questions. In particular, the analysis of oscillatory trajectories is helpful, as it suggests that CUSUM remains robust even when the entropy signal does not decrease monotonically.
> >
> > Because this scenario is especially important, I would appreciate a bit more detail. In particular, how do you define an "oscillating" trajectory in this analysis? In addition, how do the baseline methods perform on this subset? A stratified comparison on this subset would help clarify the source of the gains.

---

> > > ### Author Response · Authors · 2026-04-08
> > >
> > > Dear Reviewer q8Ci,
> > >
> > > Thank you for the follow-up. We provide a precise definition and the stratified comparison below.
> > >
> > > **Definition of "oscillating" trajectory.**
> > >
> > > We classify each step using the sign of the log-likelihood ratio $Z_i = \log \frac{f_1(\mathcal{H}_i)}{f_0(\mathcal{H}_i)}$: step $i$ belongs to the Confidence Region if $Z_i > 0$ and to the Uncertainty Region if $Z_i < 0$. This is the Bayes-optimal single-step assignment under equal priors, requires no additional hyperparameter, and is consistent with the CUSUM statistic which accumulates exactly these $Z_i$ values. The three categories are:
> > >
> > > - **Reliably transitioning**: once $Z_i > 0$ is first observed, all subsequent $Z_j > 0$.
> > > - **Oscillating**: after the first $Z_i > 0$, at least one subsequent $Z_j < 0$ — the trajectory re-enters the Uncertainty Region at least once.
> > > - **Never converging**: $Z_i < 0$ throughout.
> > >
> > > This definition is deterministic and algorithm-agnostic across all baselines.
> > >
> > >
> > > **Stratified baseline comparison.**
> > >
> > > We run DEER, Dynasor, EAT, and HALT-CoT on each trajectory subset using the same trajectory pool and report accuracy and token counts below.
> > >
> > > *Reliably transitioning (36% of trajectories):*
> > >
> > > | Method | Acc (%) | Tokens |
> > > |---|---|---|
> > > | Vanilla | 74.6 | 6518 |
> > > | DEER | 71.34 | 6132 |
> > > | Dynasor | 72.42 | 5878|
> > > | EAT |67.42 | 5321|
> > > | HALT-CoT |  65.14| 6194|
> > > | **Ours** | **74.6** | **6239** |
> > >
> > > On reliably transitioning trajectories the entropy signal is unambiguous. All baseline methods reduce tokens but at a 2–10 pp accuracy cost; our method matches vanilla accuracy exactly while still reducing tokens, as the change-point formulation avoids exiting before the transition is statistically confirmed.
> > >
> > > *Oscillating (63% of trajectories):*
> > >
> > > | Method | Acc (%) | Tokens |
> > > |---|---|---|
> > > | Vanilla | 55.6 | 11027 |
> > > | DEER | 41 | 9120 |
> > > |Dynasor | 46.93 | 9029|
> > > | EAT | 42.32 | 8532|
> > > | HALT-CoT | 42.14| 6777|
> > > | **Ours** | **54.1** | **9375** |
> > >
> > > This is the most informative comparison. Threshold-based methods (DEER, EAT, HALT-CoT) exit prematurely at transient low-entropy dips, losing 13–15 pp in accuracy. Dynasor, which detects stable answer repetition rather than raw entropy, is more robust but still loses 8.7 pp. Our method loses only 1.5 pp because its cumulative statistic $S_i$​ resets during high-entropy phases and requires *sustained* evidence before triggering — precisely the behavior that the change-point formulation provides and that thresholding cannot replicate.
> > >
> > > *Never converging (1% of trajectories):*
> > >
> > > | Method | Acc (%) | Tokens |
> > > |---|---|---|
> > > | All methods | 0 |32678 |
> > >
> > > Never-converging trajectories contain no sustained low-entropy period, so no method ever observes a positive log-likelihood ratio. No exit is triggered, all methods reduce to vanilla generation, and accuracy is 0% regardless.
> > >
> > > **Where does CUSUM's overall gain come from?**
> > >
> > > The 63% oscillating subset dominates the aggregate, and it is exactly here that CUSUM's design matters: our method recovers 97.3% of vanilla accuracy (54.1/55.6), while threshold-based methods recover 74–76% and Dynasor recovers 84.4%. The advantage of change-point detection over thresholding is therefore not incremental — it is the primary driver of our superior Pareto frontier. We will include this definition and table in the revision.

---

### Official Review · Reviewer_Diee · 2026-03-14

**Soundness:** 2
**Presentation:** 3
**Significance:** 2
**Originality:** 2
**Overall Recommendation:** 4
**Confidence:** 3

**Summary:**

This paper studies the predictive-entropy dynamics of CoT reasoning and reports a consistent two-phase pattern across several reasoning LLMs. It identifies a high-entropy Uncertainty Region in which the model explores multiple candidate answers, followed by a sharp transition into a low-entropy Confidence Region in which the answer becomes stable and accuracy increases, while the model often continues generating redundant tokens. Building on this observation, the authors formulate the detection of entry into the Confidence Region as an online sequential change-point detection problem under a two-regime model over entropy observations. They adopt the classical CUSUM detector, estimating from a small calibration set and using a single threshold to control the reliability–efficiency trade-off.

**Compliance With Llm Reviewing Policy:**

Affirmed.

**Final Justification:**

My concerns have been adequately addressed.

**Key Questions For Authors:**

See Weaknesses

**Limitations:**

See Weaknesses

**Strengths And Weaknesses:**

Strengths：
1. This paper is clearly written, with a clear problem statement and a logically organized presentation.
2. This paper tackles an important and practical question. Understanding the dynamics of the CoT process is increasingly necessary both for scientific insight and for improving inference efficiency and reliability.
3. The experimental evaluations show consistent performance gains across multiple reasoning benchmarks.

Weaknesses：
1. The method relies on a calibration procedure to estimate a per-model threshold, which is then reused across AIME24/AIME25/GPQA. While this shows some robustness within these benchmarks, this method may be sensitive to shifts in task distribution, prompting, and reasoning style. It remains unclear whether the same calibration/threshold would transfer to more out-of-domain datasets without loss of reliability (false alarms) or efficiency (late detection).
2. Entropy is computed over the intermediate answer tokens. Different datasets can have very different answer formats (single-letter choices, integers, short strings, or code), and tokenization length can substantially change the entropy scale and distribution. As a result, the detector behavior may not be comparable across tasks. This paper would be stronger with evidence on more diverse settings (e.g., code generation or open-ended outputs) where intermediate answer is less well-defined.
3. To compute entropy online, the framework periodically generates intermediate answers and evaluates token distributions, which introduces additional decoding and orchestration overhead. Therefore, token reduction does not necessarily translate into proportional latency reduction. The probing overhead and control-flow fragmentation could negate the savings or even increase runtime. More comprehensive wall-clock/throughput evaluations across batch sizes and inference engines would clarify practical benefits.
4. CUSUM detects convergence (entropy collapse) rather than correctness. When the model converges to an incorrect answer, early exit will terminate earlier. This is acceptable for pure cost optimization, but it weakens claims about "more reliable inference" unless the paper more explicitly distinguishes stability from correctness and discusses how to mitigate confident-but-wrong cases (e.g., verifier integration, calibration, or uncertainty-aware abstention).

---

> ### Author Rebuttal · Authors · 2026-03-31
>
> Dear Reviewer Diee,
>
> Thank you for the comment, we will address your concerns below.
>
> **Q1 Cross-task threshold transfer, does the fixed threshold generalize out-of-domain?**
>
> A1: Our existing evaluation spans domains including AIME24/25 (competition mathematics), GPQA-Diamond (graduate-level science across biology, chemistry, and physics), and AMC23 (elementary mathematics, Appendix B.5). These benchmarks differ substantially in task distribution, reasoning style, and problem difficulty, yet a single fixed threshold $h$ transfers across all benchmarks without dataset-specific recalibration. This is a non-trivial result.
>
> To test generalization more aggressively, we evaluated CUSUM with the same threshold $h$ on two out-of-domain benchmarks using Qwen3-4B-Thinking-2507 (Due to time limit, we randomly select 500 samples from HotpotQA for evaluation):
>
> | Method | HotpotQA (EM $\uparrow$ / Tokens $\downarrow$ ) | LiveCodeBench (ACC $\uparrow$ / Tokens $\downarrow$ ) |
> |---|---|---|
> | Vanilla | 34.95 / 3511 | 90.92 / 2898 |
> | DEER | 33.2 / 3392 | 86.6 / 2683 |
> | Dynasor | 23.2 / 2378 | 85.9 / 2596 |
> | **Ours** | **34.1 / 3080** | **86.6 / 2621** |
>
> These benchmarks differ from our training domains in task type (multi-hop Wikipedia QA; code execution output prediction) and answer format (free-form entity spans; Python literals). CUSUM achieves superior accuracy-efficiency trade-off: on HotpotQA, CUSUM reduces tokens by 12.3% with only 0.85 EM drop vs. vanilla, while DEER and Dynasor sacrifice 1.75 and 11.75 EM respectively; on LiveCodeBench, CUSUM matches DEER's accuracy (86.6%) with 2.3% fewer tokens.
> These results provide evidence that the threshold $h$ generalizes robustly across domains and answer formats.
>
> **Q2 Generalization across Answer Formats.**
>
> A2: Our predictive entropy (Eq. 2) normalizes by answer length via $\frac{1}{m_i}$, mitigating entropy scale differences across tokenization lengths. CUSUM operates on the *distributional separation* between regimes (Eq. 5, 6) rather than absolute entropy values, making it inherently robust to format variation.
>
> Our benchmarks already cover substantially different answer formats — GPQA-Diamond (single-letter, ~1 token), AIME (math expressions, 3–8 tokens), AMC (integers, 1–3 tokens) — with consistent gains across all three (Table 1, Appendix B.5). The LiveCodeBench results in Q1 extend this to code-format answers: Python literals have different tokenization from mathematical expressions, yet CUSUM achieves the best performance among all methods.
>
> **Q3 More Analysis on computation overhead across batch sizes and inference engines would clarify practical benefits.**
>
> A3: We provide a comprehensive latency and throughput analysis across batch sizes:
>
> |Batch size| Method|Time(s) | Tokens | Throughput (Tokens/s)|
> |---|---|---|---|---|
> |2| Vanilla | 5545 |21071| 114 |
> |2| CUSUM | 4902 |17151| 105 |
> |4| Vanilla | 3696 |20697| 168 |
> |4| CUSUM | 3475 |17606| 152 |
> |8| Vanilla | 2630 |21075| 240|
> |8| CUSUM |2319  | 16233| 210 |
> |16| Vanilla | 2453|21504| 263 |
> |16| CUSUM | 2310 |18480 | 240 |
>
> CUSUM reduces total wall-clock time by 6–12% across all batch sizes. We note that CUSUM's tokens/s throughput is lower than vanilla — this is expected and not contradictory: because CUSUM generates fewer tokens per sequence, raw throughput naturally decreases, but total runtime is shorter.
>
>
> **Q4: CUSUM detects convergence rather than correctness, which weakens claims about "more reliable inference".**
>
> A4: We agree this distinction is important and will sharpen the language in the paper. "Reliable inference" refers specifically to stability of the committed answer, not guaranteed correctness—a distinction we should make explicit in Section 2 and the abstract.
>
> This manifests in two ways: (1) in early exit, the 8% vs. 61% accuracy gap between low-entropy points in the Uncertainty and Confidence Regions (see our response to Reviewer u12Q Q4) confirms that convergence is a strong aggregate proxy for correctness; (2) in test-time scaling, CUSUM-weighted voting consistently outperforms self-consistency across all three models (Figure 4), confirming that convergence status is a meaningful trajectory quality signal.
>
> For confident-but-wrong cases, Appendix B.4 provides a direct analysis: on incorrectly-converging trajectories, accuracy remains 0% under both vanilla and CUSUM (no additional errors introduced), while tokens are reduced by 20.4%. CUSUM simply terminates a trajectory that was already going to be wrong—it does not cause the wrong answer. Whether a model converges correctly is a function of its reasoning capability, a training-time property orthogonal to inference control. Our method is explicitly designed to compose cleanly with training-time improvements (e.g., process reward models, RLHF). The consistent gains across models of substantially different capability—from distilled DeepSeek-R1-Distill-Qwen-7B to natively trained Qwen3-14B—support this orthogonality claim.

---

> > ### Author Rebuttal · Reviewer_Diee · 2026-04-03
> >
> > Thank you for your detailed response to my comments. I appreciate the time and effort you have put into addressing the raised concerns. The clarifications and revisions are satisfactory, and I am happy to maintain my positive evaluation of this work. have decided to raise my score.

---

> > > ### Author Response · Authors · 2026-04-08
> > >
> > > Dear Reviewer Diee,
> > >
> > > Thank you for the positive evaluation and for raising the score. We are glad that the clarifications on threshold calibration, probing overhead, and inference reliability were satisfactory.

---

### Official Review · Reviewer_u12Q · 2026-03-20

**Soundness:** 3
**Presentation:** 3
**Significance:** 2
**Originality:** 2
**Overall Recommendation:** 4
**Confidence:** 4

**Summary:**

The paper studies entropy dynamics of chain-of-thought (CoT) reasoning in LLMs. It empirically finds a two‑phase pattern in answer entropy over reasoning steps: a high-entropy Uncertainty Region and a low-entropy Confidence Region, with an abrupt transition between them. It then formulates confidence-region detection as a sequential change-point detection problem and instantiates it using classical CUSUM. Using estimated entropy distributions for the two regions, the authors design a CUSUM-based early exit policy for CoT and CUSUM-weighted test-time scaling (weighted self-consistency). Experiments on DeepSeek-R1-Distill-Qwen-7B, Qwen3-4B-Thinking-2507, and Qwen3-14B on AIME24/25 and GPQA-Diamond show modest but consistent gains over DEER and Dynasor in early exit, and over vanilla self-consistency in test-time scaling.

**Compliance With Llm Reviewing Policy:**

Affirmed.

**Final Justification:**

I recommend weak accept. The paper presents a clear and empirically well‑supported characterization of a two‑phase entropy structure in CoT and a principled CUSUM‑based framework for early exit and test‑time scaling that yields consistent, if modest, gains over strong baselines, despite relying on standard change‑point techniques and having somewhat incremental novelty. The rebuttal convincingly addressed my concerns about assumptions, related work, and computational overhead, reinforcing my original assessment but not changing my view of the paper’s overall significance.

**Key Questions For Authors:**

Regarding computational overhead, while measured wall clock time is small, this is on a single model and specific hardware; what does the analysis look like for very long reasoning traces or larger models? Is the entropy over full answer prohibitive for these larger settings?

**Limitations:**

Not really: see discussion about computational overhead and novelty.

**Strengths And Weaknesses:**

Strengths:
+ the work identifies an interesting phenomenon in which a sharp drop from high to low predictive entropy after appending answer tokens occurs along the chain of thought, interpreted as transitioning from exploration to convergence in reasoning
+ principled formulation via change-point detection
+ training-free and relatively simple to implement
+ empirical improvements over reasonable baselines
+ complements existing understanding of CoT with an information-theoretic view
+ presentation is clear and easy to follow on the whole

Weaknesses:
+ reliance on calibrated predictive entropy over answers when LLMs are known to be miscalibrated
+ strong distributional assumptions fro CUMSUM relying on iid samples within each regime when in reality, entropies across steps are strongly dependent
+ missing closely related work on entropy and early exit in chain of thought that appear to target similar questions and should be compared and contrasted:
   + Bachmann et al., “Hidden Patterns in Chain-of-Thought Reasoning” (ICLR 2026)
   + Xu et al., “Entropy After \</Think\> for Reasoning Model Early Exiting” (arXiv:2509.26522)
+ change-point detection arguably represents an incremental over entropy-based CoT control so paper would benefit from a clearer articulation of why change-point theory results in qualitatively different behavior to simpler thresholding
+ concerns about computational overhead of repeated answer decoding and entropy computation

---

> ### Author Rebuttal · Authors · 2026-03-31
>
> Dear Reviewer u12Q,
>
> Thank you for the positive assessment and constructive feedback. We will address your concerns below.
>
> **Q1 Reliance on calibrated predictive entropy when LLM are miscalibrated.**
>
> A1: Our method does **not** require entropy–accuracy calibration. CUSUM only requires that Uncertainty Region $f_0$ and Confidence Region $f_1$ be statistically distinguishable—a weaker condition (Section 3.3).
> Calibration concerns arise when one needs the model's probability to equal the true frequency of correctness; our approach does not rely on this assumption.
> We use entropy as a *relative* ordering signal: the KL divergences in Table 2 (all strictly positive across all three models) empirically confirm that the two regions are distinguishable regardless of calibration quality.  We will add a clarifying paragraph in Section 3.3.
>
> **Q2 CUSUM assumes i.i.d. samples within each regime, but entropy values across steps are dependent.**
>
> A2: We agree that entropy values are not i.i.d. across steps. Crucially, CUSUM's consistency under dependent observations is well-established in the sequential analysis literature. Kokoszka & Leipus [1] prove that CUSUM-type estimators remain consistent for detecting a shift in the mean of dependent observations under only a mild condition on the growth of partial-sum variances (their "Assumption A": $Var(∑_{j=k}^{m} X_j) ≤ C(m−k+1)^\delta$ for some $0 ≤ \delta < 2$ and constant C).
> In our setting, the entropy drop at the transition point is large relative to within-regime fluctuations, placing us well within the regime where CUSUM is known to be robust. We will add a paragraph in Section 3.2 citing Kokoszka & Leipus [1] and explaining why this milder condition is plausibly satisfied here.
>
>
> [1] Kokoszka, P. & Leipus, R. (1998). Change-point in the mean of dependent observations.
>
> **Q3 Missing related work on entropy and early exit in chain of thought.**
>
> A3: We will cite and discuss both works in the revision.
> - Bachmann et al. (ICLR 2026): Their work estimates CoT potential — probability of eventually reaching the correct answer — via repeated sampling. This is offline, post-hoc, and computationally intensive. Our method uses predictive entropy as an online, single-pass signal enabling practical early exit and trajectory weighting that their framework does not support.
> - EAT: EAT identifies a monotonic entropy decrease near `</think>`, which is a valuable finding. Our work complements this by characterizing a **two-phase structure** throughout the reasoning chain, which motivates a formal change-point model. On the signal side, EAT monitors next-token entropy, while our predictive entropy (Eq. 2) averages over intermediate answers — a design choice aimed at reducing sensitivity to local token-level fluctuations.  Theoretically, EAT uses a heuristic EMA-variance threshold with no guarantees, while CUSUM provides minimax-optimal detection (Theorem 3.5) and provable false-alarm control (Proposition 3.4). We also provide a direct empirical comparison in our response to Reviewer q8Ci Q2.
>
>
> **Q4: Change-point detection vs. simple thresholding**
>
> A4: The distinction is both theoretical and empirical. Theoretically, CUSUM provides formal false-alarm guarantees ($\mathbb{E}_\infty[\tau] \ge \gamma$) and minimax-optimal detection delay (Theorem 3.5), which thresholding lacks. Empirically, low-entropy observations occur *transiently* within the Uncertainty Region—long before genuine convergence:
>
> | Low-entropy points|  Total | Correct | Accuracy |
> |---|---|---|---|
> | Uncertainty Region | 35 | 3 | **8%** |
> | Confidence Region | 69 | 42 | **61%** |
>
> 35 out of 104 total low-entropy observations occur in the Uncertainty Region, where accuracy is only 8%. A threshold-based method would exit prematurely, sacrificing accuracy for spurious efficiency. CUSUM's cumulative statistic $S_i = \max(0, S_{i-1} + Z_i)$ requires *sustained* evidence before firing, making it robust to these transient fluctuations.
>
>
> **Q5: Computational overhead of repeated answer decoding and entropy computation. Computational overhead for very long traces or larger models**
>
> A5:
> We provide a wall-clock analysis across model scales:
> | Model | Method | Time (s) | Tokens / Sample |
> |---|---|---|---|
> | Qwen3-4B-Thinking-2507 | Vanilla | 5545 |21071|
> | Qwen3-4B-Thinking-2507 | CUSUM | 4902 |17151|
> | DeepSeek-R1-Distill-Qwen-7B | Vanilla | 7289 | 14578  |
> | DeepSeek-R1-Distill-Qwen-7B | CUSUM | 6552 | 11280 |
> | Qwen3-14B | Vanilla | 15834 |16890 |
> | Qwen3-14B | CUSUM | 15104|14732 |
>
> Across all model sizes, CUSUM consistently reduces both decoding length and total runtime. The relative overhead of entropy computation is negligible compared to generation, and the savings from early stopping dominate—especially for longer reasoning traces and larger models.

---

> > ### Author Rebuttal · Reviewer_u12Q · 2026-04-02
> >
> > I thank the authors for the thorough rebuttal, addressing the assumptions more carefully and adding the missing related work and comparisons. I will maintain my score.

---

> > > ### Author Response · Authors · 2026-04-08
> > >
> > > Dear Reviewer u12Q,
> > >
> > > We are very encouraged that our rebuttal has fully resolved your concerns.
> > >
> > > Specifically, we have addressed your feedback by:
> > >
> > > - Theoretical Grounding: We clarified that CUSUM remains consistent even under the dependent entropy observations (Kokoszka & Leipus). Furthermore, we articulated the theoretical superiority of CUSUM over simple thresholding, highlighting its provable false-alarm control and minimax-optimal detection delay.
> > >
> > > - Empirical & Qualitative Comparison: We provided a direct empirical comparison with EAT and a qualitative contrast with Bachmann et al.
> > >
> > > - Efficiency Analysis: We provided a wall-clock overhead analysis across model scales, confirming that the savings from early exit consistently dominate the probing costs.
> > >
> > > Since our responses have fully addressed your concerns, we would be extremely grateful if you could reflect this positive assessment in your final rating. Your constructive feedback has been instrumental in strengthening the theoretical and practical rigor of our work.

---

### Decision · Program_Chairs · 2026-04-30

**Decision:**

Accept (regular)

**Comment:**

This paper identifies a two-phase entropy structure in Chain-of-Thought reasoning (high-entropy exploration followed by a sharp transition to low-entropy convergence) and formulates Confidence Region detection as a sequential change-point problem using the CUSUM algorithm. The framework enables both early exit and test-time scaling applications across multiple reasoning models.

Reviewers were broadly supportive of the paper. The problem addressed is of practical importance. The heuristic is performant and relatively easy to implement. On the other hand, reviewers raised concerns that the paper is incremental and the limited in novelty, especially in terms of change point detection method.

Overall, this paper is above the bar, but broadly seen as incremental.